# Reinforcement Learning Control of a Physical Robot Device for Assisted Human Walking without a Simulator

**Junmin Zhong** [1]  **Emiliano Quinones Yumbla** [2]  **Seyed Yousef Soltanian** [2]  **Ruofan Wu** [1]  **Wenlong Zhang** [2 3]
**Jennie Si** [1 3]

## Abstract

This study presents an innovative reinforcement learning (RL) control approach to facilitate soft exosuit-assisted human walking. Our goal is to address the ongoing challenges in developing reliable RL-based methods for controlling physical devices. To overcome key obstacles—such as limited data, the absence of a simulator for human-robot interaction during walking, the need for low computational overhead in real-time deployment, and the demand for rapid adaptation to achieve personalized control while ensuring human safety—we propose an online *Adaptation* from an offline *Imitating* Expert *Policy* (AIP) approach. Our offline learning mimics human expert actions through real human walking demonstrations without robot assistance. The resulted policy is then used to initialize online actor-critic learning, the goal of which is to optimally personalize robot assistance. In addition to being fast and robust, our online RL method also posses important properties such as learning convergence, dynamic stability, and solution optimality. We have successfully demonstrated our simple and robust framework for safe robot control on all five tested human participants, without selectively presenting results. The qualitative performance guarantees provided by our online RL, along with the consistent experimental validation of AIP control, represent the first demonstration of online adaptation for softsuit control personalization and serve as important evidence for the use of online RL in controlling a physical device to solve a real-life problem.

---

[1]School of Electrical, Computer and Energy Engineering, Arizona State University, Tempe, Arizona. [2]School of Manufacturing Systems and Networks, Arizona State University, Mesa, Arizona. [3]Corresponding Authors. Correspondence to: Jennie Si <si@asu.edu>, Wenlong Zhang <wenlong.zhang@asu.edu>.

*Proceedings of the 42$^{nd}$ International Conference on Machine Learning*, Vancouver, Canada. PMLR 267, 2025. Copyright 2025 by the author(s).

## 1. Introduction

**Goal of this study.** Wearable robots such as rigid exoskeletons and soft exosuits have been extensively researched and have shown great promise for gait rehabilitation (Rodríguez-Fernández et al., 2021) and for assisting human walking to reduce physical efforts (Collins et al., 2015). Unlike rigid exoskeletons, soft, garment-like devices made from materials like silicone elastomers and fabrics provide a more comfortable, safer, and adaptable user experience (Granberry et al., 2017; Save et al., 2025; Yumbla et al., 2021). Yet, effectively controlling the wearable robots to seamlessly work with human users in locomotion tasks remain a major challenge. This may be why deployment of the promising wearable technology still have limited success in real-world deployment. Reinforcement learning points to two potential solutions: sim-to-real approach or direct design in the real physical environment. However, devising a near-perfect simulator for human-soft exosuit walking is exceptionally costly if at all possible given the unpredictable human behavior while wearing a foreign robotic device, and the ubiquitous presence of noise, delay, and other uncertainty in the environment. **Challenges of controlling a soft wearable exosuit.** First off, soft inflatable exosuits lack a robust model of their dynamics (Polygerinos et al., 2015), not to mention modeling the interaction dynamics between the user and the robot, a necessary step in building a high fidelity simulator. These unique challenges stem from that the pneumatic dynamics of the soft inflatable actuators (Joshi & Paik, 2021) are complicated in part by the nonlinear nature of soft actuators due to material properties and design geometry. The fabric-based actuators result in highly compliant behavior that enables high levels of deformations (Hasan et al., 2022). The manufacturing process of the actuators also introduces significant variations and uncertainties (Joshi & Paik, 2021). Further wear and tear of the fabric only makes problem more complicated. Lacking a reliable model or simulator of the soft robot has made controlling a soft inflatable exosuit more complex than a traditional rigid exoskeleton (Polygerinos et al., 2015). Unlike rigid exoskeletons where the assistive torque is determined by motor actuators and can be directly used as a control parameter, for soft inflatables, the torque is generated from two collaborative sources: the human and

the exosuit, which is nearly impossible to quantify. Additionally, the inflation/deflation of the actuators typically introduces longer actuation delays than motor-actuated exoskeleton, a factor that potentially reduces stability margins in control system design. **Contributions of this study.** Our approach of online *Adaptation* from an offline *Imitating* expert *Policy* (AIP) provides a holistic framework that enables efficient, effective, and direct physical device control in order to overcome the challenges of lacking a simulator, limited data, and robustness in real time deployment. Specifically, 1) instead of an algorithm-centric approach that has been mostly developed and deployed based on extensive simulations or vast amount of data in successful offline-to-online robot RL control (Levine et al., 2020; Kumar et al., 2020; Lee et al., 2022) we rely on a data-centric approach to account for environmental noise, delay, and other uncertainties. 2) The actor-critic online adaptation approach has shown its capability of successfully addressing distribution shift as we adapt offline learned policy to individual new users. Validations are provided by real experiments involving human users walking with the soft exosuit. Results successfully show that soft exosuit control has assisted human normative walking with reduced human effort. 3) Additionally, we provide qualitative online learning performance assurances such as learning convergence, dynamic stability and human safety, and solution optimality. The qualitative performance guarantees, along with the consistent experimental validation of AIP control, represent the first demonstration of online adaptation for softsuit control personalization and serve as important evidence for the use of online RL in controlling a physical device to solve a real-life problem.

## 2. Related Work

**Control of Soft Exosuit.** A fundamental control challenge with wearable devices is modeling the interaction dynamics between the user and the robot for optimal coordination (Polygerinos et al., 2015; Nesler et al., 2018; O'Neill et al., 2022; Liu et al., 2022; Wu et al., 2022), and additional challenges as discussed in Appendix B. Several studies have achieved successful coordination through human-in-the-loop optimization methods. In Siviy et al. 2020, offline optimization of a cable-driven ankle exosuit is performed to generate the assistive torque profile. In Ding et al. 2018, the authors perform human-in-the-loop optimization through a Bayesian optimization to identify the peak and offset timing of hip extension assistance with a cable-driven hip exosuit. In Kim et al. 2019b, the authors advance this framework by coupling Bayesian optimization with a Kalman filter metabolic estimator to deliver plantar flexion assistance to the ankle with a cable-driven ankle exosuit. In Li et al. 2022a, the authors developed a hierarchical human-in-the-loop controller of a cable-driven exosuit for impedance

adaptation to different terrains. An offline cable control parameter optimization was developed in Li et al. 2022b, which relies on an impedance model based on the geometric relationship of ankle joint. While these studies have achieved coordination between the robot and the user, they have a strong prerequisite that the wearable robot possesses a robust dynamical model. They also lack the ability of online tuning and personalizing for different users. Therefore, new innovations are needed for those wearable devices ready for real life deployment.

**Offline-to-online RL** is an effective and practical approach for robot control. Extensive research effort has been put into addressing the distribution shift from offline to online. The issue has been explored through various approaches. Zhang et al. 2023 introduces policy expansion which expand the policy with another learnt policy and implements a Boltzmann action selection strategy. Several studies (Lee et al., 2022; Zhao et al., 2023a;b) utilize an ensemble of pessimistic value functions to mitigate distributional shift. Zhong et al. 2024 develops a sim-to-real application on gantry crane. Nakamoto et al. 2024 proposes training an additional value function to address over-conservatism. Li et al. 2023 introduces a policy regularization term for trust-region-style updates. Lei et al. 2023 uses an on-policy optimization strategy unify offline and online training without extra regularization. **However,** nearly all of the current offline-to-online RL approaches rely on extensive simulations or a vast amount of offline data, and none of them have demonstrated directly learning from physical environment using only limited data for an offline policy (Levine et al., 2020; Kumar et al., 2020). Furthermore, almost all the demonstrated robot control applications are under structured environment such as those in warehouse settings or performing specific tasks.

The PPO and SAC methods have been widely used in online robotic learning, again, relying heavily on extensive simulations for under structured environment learning. Both algorithms have shown great ability to learn stochastic policies by parameterizing a probability distribution over actions. These distribution-based policies facilitate exploration by sampling actions during training (Haarnoja et al., 2018; Schulman et al., 2017). **However**, applying PPO and SAC in soft exosuit robot that does not have a simulator or a dynamic model of human-robot interaction, presents true challenges. **Additionally**, in wearable robotics with human in the loop, user comfort and safety are of great importance. Human participants become frustrated if the robot operates erratically (Dani et al., 2020). This necessitates exploration of using deterministic control policies.

**Data-centric approach in offline imitation learning (IL).** IL is a natural and effective part of reinforcement learning (RL) to device a reasonable initial policy (Taylor et al.,

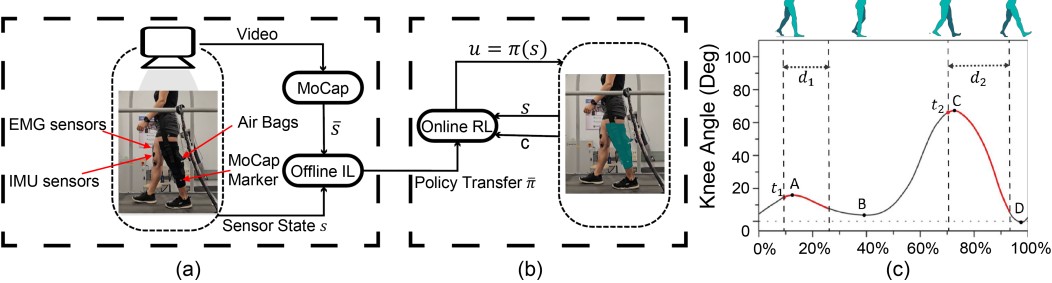

Figure 1. (a): Offline imitation learning using normative human walking data with ground truth provided by MoCap. The learned policy is then used to initialize online RL. (b): Online, personalized RL control of the soft exosuit to achieve human-robot normative walking while minimizing human effort measured by EMG activity. Sensor data are acquired via IMU for real-time control. (c): Knee angle profile of a complete gait cycle (in %) with the four gait phases as shown.

2011). To address the issue of distribution shift when applied in online environments (Ross et al., 2011; Spencer et al., 2021), two typical approaches have been explored. **Algorithm-centric approaches** aim to learn robust policies by imposing task-specific assumptions based on specific characteristics of the task (Galashov et al., 2022; Guhur et al., 2023; James & Davison, 2022), or acquiring additional data to model environment dynamics for the agent to return to in-distribution states (Englert et al., 2013; Qi et al., 2022). Some approaches enhance action representation such as using Gaussian or mixture models to capture all expert actions (Chi et al., 2023; Mandlekar et al., 2021). Others reduce the task length by employing temporal abstraction of the action spaces (Shridhar et al., 2023; Zhao et al., 2023c). **Data-centric approaches** prioritize data quality, primarily aiming to maximize state diversity. Numerous studies focus on modifying data collection processes to expose the expert to a diverse set of state transitions through shared control (Cui et al., 2019; Kelly et al., 2019; Ross et al., 2011). Some methods allow human intervention to correct robot behavior when necessary (Gandhi et al., 2023; Mandlekar et al., 2020). Active learning guides data collection toward more informative samples by prioritizing questions that maximize information gain while minimizing the difficulty of selecting queries(Bıyık et al., 2019; Cui & Niekum, 2018). However typical data-centric techniques, such as collecting more data, diversifying state transition, actively learning human walking dynamics, or human intervention of natural and normative walking, are entirely unfeasible for easy to understand reasons. We focus on improving **data quality** with two specific considerations: 1) to avoid long data collection processes, and 2) effectively deal with environmental noise to address our unique problem challenge. With such a data-centric framework, we expect to improve action divergence (Belkhale et al., 2024) between the learned policy and the demonstration policy, thereby to improve task success rates.

## 3. Method

Our online *Adaptation* from an offline *Imitating* expert *Policy* (AIP) procedure consists of two main phases. First, we develop a data-centric offline learning approach by employing IL. We aim to capture a baseline normative walking policy by focusing on improving data quality and thus the quality of the initial policy for online adaptation. Second, to personalize RL controller for new users online, we use an actor-critic online learning method, the direct heuristic dynamic programming (dHDP) (Si & Wang, 2001), which is a deterministic policy gradient method. It is based on the idea that the actor adjusts the policy in the direction of the action-value gradient, and the critic updates the action-value function. The idea can also be found in NFQCA (Hafner & Riedmiller, 2011), which in turn, in DPG (Silver et al., 2014) and DDPG (Lillicrap et al., 2015). A more detailed discussion on dHDP and its performance evaluation in comparison to DDPG, as well as its several significant applications in complex and realistic engineering systems can be found in Appendix B. Ultimately, due to the holistic considerations that define the AIP method, it is one of the first realizations that does not rely on a simulator or vast amount of offline data to train an offline policy and successfully adapt it online for new users.

### 3.1. Physical Setup

Refer to Figure 1, the AIP solution involves two main phases, offline imitation learning (Figure 1.a) and online personalized RL (Figure 1.b). In both phases, a participant walks on the treadmill at a constant speed of 1 m/s. The inertial measurement unit (IMU) sensors collect kinematic data while the electromyography (EMG) sensors measure muscle activity simultaneously. A Motion Capture (MoCap) System, which provides ground truth measurement of the human joint motion, is time synced with the IMU and EMG sensors in the offline phase and the ground truth walking profiles were used to train an offline human policy as an initial policy

for online training. The complete anthropometric data of the subjects and IRB can be seen in Appendix A and details on the placement of the soft inflatable exosuit, its manufacturing, wearable sensors, etc. can be found in Appendix D. Experiment protocol can be found in Appendix E.

## 3.2. State and Control Variables

An analysis of knee joint kinematics reveals two critical regions associated with knee stance extension (from point A to B) and swing extension (from point C to D) during a gait cycle (Figure 1.c). The four extrema mark the transition from one gait phase to another: marker A is the maximum knee flexion during mid stance, B the maximum knee extension during terminal stance phase, C the maximum knee flexion during mid swing phase, and D the maximum knee extension adjacent to heel strike. These transition points and their related characteristics are therefore considered for inclusion in the state representation from the human walking profile. Specifically, the state variables include the peak knee flexion angle at point C as denoted by $\theta_f$, the time instances $t_A$ and $t_C$ of peak knee flexion at the stance and swing phases, respectively, the duration of the stance phase (between point A and point B) defined as $d_A = t_B - t_A$, and the duration of the swing phase (between point C and point D) defined as $d_C = t_D - t_C$. Thus, we define the state variable as follows:

$$s = [t_A, d_A, t_C, d_C, \theta_f]^T. \qquad (1)$$

Unlike rigid exoskeletons where the torque is generated by electrical motors and can be directly used as a control parameter, for soft inflatable actuators, the amount of assistive torque is determined by both human knee torque and the actuator pressure, the two collaborative sources. It is therefore not feasible to use torque directly as the control variable for the exosuit. Instead, only properly timed inflation and deflation of the exosuit will provide the necessary and optimal assistance to the human user (Figure 1.c). On the contrary, if the exosuit is not properly operated, it may cause discomfort or even injury to the human user. Toward this end, the RL controller must determine the optimal timings to operate the exosuit and these control parameters are:

$$u = [t_1, d_1, t_2, d_2]^T, \qquad (2)$$

where $t_1$ is the onset timing of inflation of the exosuit to assist stance extension, the duration for which the air pressure is maintained during this phase is $d_1$. Similarly, $t_2$ represents the onset timing of inflation of the exosuit to assist swing flexion, and the corresponding duration for maintaining air pressure during this phase is given by $d_2$. As it takes time for the exosuit to inflate and deflate, it is expected that an optimal RL controller should successfully learn the optimal timings of $t_1$ and $t_2$, which are expected to be close to or ahead of the maximum flexion timings $t_A$

and $t_C$. By precisely adjusting these timings and durations, the RL controller ensures that the exosuit provides optimal assistance to the user's knee movement, enhancing overall gait efficiency and reducing muscle effort quantified by EMG measurement (EMG Effort). Note that, it is natural to maintain consistency in the state and action spaces during both offline and online learning.

## 3.3. Safety Constraints

To ensure human participants walk continuously and safely, we consider several safety constrains: 1) the actuator pressure is limited to 206.8 kpa; 2) the control timings and inflation/deflation durations are constrained by taking reference of those during participant's normative walking as shown in Table 4, which are within realistic ranges (Zhang et al., 2020b). These physical constraints help prevent significant misalignment between controller timing and the respective gait phase during human walking. Without these constraints, it may trigger soft actuator deployment and cause discomfort or injury to the user; and 3) the online training objective is set for the control timings to approach those during normative walking, and thus in a safe state. Details of the safety constraints and their physical representations are in Appendix C. Additionally, we provide a theoretical performance analysis of the online learning process to ensure learning convergence, optimal timing solution, and human-robot interaction dynamic stability under reasonable conditions and within these safety constraints (Appendix F).

## 3.4. Offline Human Normative Walking Policy

We take a different perspective of not focusing on developing another offline method, but instead, focusing on how to improve data quality to make offline imitation learning more effective in developing a good quality offline policy. By doing so, we aim to demonstrate the generalizability and data efficiency of our method by ONLY collecting offline walking data from a SINGLE participant (participant 1) with $N = 150$. Detailed information about the offline data collection process can be found in Appendix E. Our imitation learning approach utilizes Behaviour Cloning (BC) (Torabi et al., 2018; Bain & Sammut, 1995; Daftry et al., 2017) to derive an effective imitation policy based on data $\overline{\mathcal{D}} = \{\overline{s}(k)|, k = 1, 2, ..., N\}$, obtained from normative walking demonstrations under natural walking condition of a human participant, where $N$ represents the total number of gait cycles over which the state variable data is collected from the MoCap system.

A good offline normative walking policy should serve the following two purposes. First, it provides a reasonable initial policy for online tuning tailored for individual users while both offline and online learning are subject to simi-

lar environmental uncertainties such as sensor and actuator noise. As such, this offline policy helps online policy tuning to be kept in a reasonable and meaningful range. Second, this offline learned policy should capture key human locomotion characteristics even under intra- and inter-person variations (Zhang et al., 2020a; Ahn & Hogan, 2012) as human locomotion (such as knee angle) exhibits similar patterns as shown in Figure (1.c).

**Improve data quality.** Based on most recent results that data-centric approaches have greater impact than algorithm-centric approaches on the effectiveness of imitation learning (Belkhale et al., 2024), we aim to improve data quality and expect that to be especially effective in addressing our unique challenges associated with the human-robot system under study. We therefore propose a reducing intra-person and inter-person variation (RIIV) method to improve measured data quality as it is likely to be the one and the most effective and efficient approach. As a result, we compare our RIIV with a common benchmark approach that normalizes the raw measurements of state variables into the [-1,1] range.

Specifically, for each gait cycle of length $T$, let the original measurements of each state variable of $s$, as in Equation (1), be denoted by $\zeta$. The following computations are performed component-wise for each of the state variables ($j = 1, 2, ..., 5$).

1) The benchmark DIRECT method normalizes the raw sensor measurements ($\zeta$) of states into [-1,1] by the following procedure,

$$s = 2\left(\frac{\zeta - min(\zeta)}{max(\zeta) - min(\zeta)}\right) - 1. \tag{3}$$

2) The RIIV method.

The first step of RIIV reduces intra-person step length variations by converting gait timing from actual time into gait percentage by normalizing over a gait cycle $T$, that is

$$\xi = \frac{\zeta}{T}. \tag{4}$$

The second step reduces inter-person variation by transferring state variables into the range of $[-1, 1]$,

$$s = 2\left(\frac{\xi - \inf(\xi)}{\sup(\xi) - \inf(\xi)}\right) - 1, \tag{5}$$

where the values of $\inf(\xi)$ and $\sup(\xi)$ are from established studies of biomechanics literature (Zhang et al., 2020b), which is shown in Table 6 in Appendix C.

**Imitation policy.** Once real time measurements for offline policy training are obtained during normative human walking, BC is utilized to train an offline imitation human walking policy $\pi(s_k)$, which maps human state from IMU sensors to control timings and durations of normative walking with the ground truth provided by MoCap, namely, $\overline{\pi} = \{\overline{t}_A, \overline{d}_A, \overline{t}_C, \overline{d}_C\} \in \overline{\mathcal{D}}$. We use "action divergence" to measure offline cost $\overline{c}_k$ in BC learning,

$$\overline{c}_k = \frac{1}{2}(\pi(s_k) - \overline{\pi}_k)^2. \tag{6}$$

Therefore the actor with policy parameter ($\phi$) minimizes a supervised loss as:

$$L(\phi) = \frac{1}{N}\sum_{k=0}^{N} \overline{c}_k. \tag{7}$$

which is the distance between the RL policy and that used in human demonstration.

### 3.5. Online, Personalized Soft Exosuit Control

Personalizing soft robot control for individuals face the following offline to online learning **challenges**: 1) Data out of distribution (OOD) due to inter- and intra-human variance; 2) Limited availability of human walking data; and 3) Hardware limitations including communication delays, sensor noise, and significant delay in actuation. 4) Human acceptance of the physical device which is directly correlated with human trust in the device and comfort when walking under robot control. Our solution relies on a good data quality improvement procedure and an efficient online reinforcement learning algorithm, the dHDP which can sccessfully address the above OOD problems as shown in Appendix G.

The dHDP actor-critic learning has demonstrated online learning convergence under limited data conditions empirically in several wearable robotics applications (Wen et al., 2016a;b; 2017a; 2019). Additionally, dHDP with expeirence replay and target network has shown compatible performance to DDPG in many DMC robotic control problems (Wu et al., 2024). Along with nice theoretical properties of dHDP (Sectoin 3.6) and practical successes in previous studies, we employ dHDP in our current study to learn the exosuit control solution with 5-dim state space and 4-dim action space.

The goal of online learning control of the soft exosuit is to minimize the user's muscle effort by improving the effectiveness of the exosuit assistance. We thus consider two necessary performance metrics: gait normalcy thus safety constraint, and muscle effort. the online learning objective is therefore to minimize the overall cost over policy $\pi$, defined as follows:

$$Q^\pi(s_k, u_k) = \mathbb{E}[\sum_{t=k}^{\infty} \gamma^{t-k} c_t | s_k, u_k], \tag{8}$$

where $s_k \sim p(\cdot \mid s_{k-1}, u_{k-1})$, $u_k = \pi(s_k)$, $c_k = c(s_k, u_k)$ is the stage cost, and the discount factor $0 <$

$\gamma < 1$. The stage cost $c_k$ in the above is formulated to take into consideration of two important performance measures in online learning of personalized optimal policy to achieve robot-assisted normative walking with reduced energy expenditure.

**First**, we embed normative walking and safety constraint $\epsilon^s$ as one of the important performance considerations in the performance index (Equation 10). Specifically, $\epsilon^s = (s - \tilde{s})^2$, where target state $\tilde{s}$ is defined in Appendix C, and is extracted from the offline normative walking profile using MoCap data as in Table 4. Additionally, $\epsilon^s$ is bounded within safety constraint provided in Table 5. It ensures that the subject does not deviate significantly from the target, thereby preventing potential falls or discomfort. The **second** consideration of reducing human energy expenditure is reflected by reduced muscle activity, which is measured per gait cycle, namely, the EMG effort $\epsilon^e$ is determined by

$$\epsilon^e = \frac{1}{2}(\sum_{t=0}^{T} f_E(t))^2 \qquad (9)$$

which $f_E(t)$ is the EMG sensor value at time $t$ of a gait and $\sum_{t=0}^{T} f_E(t)$ simulates the integral of the EMG signal under a complete gait cycle of length $T$.

We thus have the stage cost $c_k$ formulated by balancing the reduction of EMG effort and adherence to state error tolerance and safety constraints, and it is consequently used in formulating the total cost in Equation (8):

$$c_k = \epsilon^s + \epsilon^e. \qquad (10)$$

The dHDP is then used to provide online learning of a personalized optimal policy for individual users. Further details about dHDP, its actor and critic network realizations, and its implementation can be found in Appendices E and F.

### 3.6. Qualitative Properties of the Learning Process and Control Performance Associated with dHDP Online Learning

We provide a theoretical analysis to characterize properties of the learning process and the control performance, specifically those related to learning convergence, solution optimality, and control system stability as a result of online dHDP learning initialized by an offline policy obtained via imitation learning. Details are provided in Appendix F.

## 4. Results and Analysis

This study of directly learning to control a physical device, the robotic exosuit, to assist normative human walking aims at exploring the feasibility of RL in achieving stable and efficient learning without a simulator. As a result, we have shown promising first steps in addressing key challenges of RL control for real life applications. Furthermore, our AIP as a data-centric, offline to online approach reveals its practical usefulness to address environment uncertainty due to variations in human, sensor and actuator noise and delay that are unavoidable in real physical environments. **Videos of the experiments can be seen in supplemental files**.

**Performance Criteria.** The results reported in this study were based on the following performance metrics: 1) The stage cost as shown in Equation 10 to reflect online learning performance; 2) Peak knee error as a kinematic measure of normative walking and also to reflect walking safety; 3) EMG activity (Equation 9) which reflects human effort during walking; 4) Time to convergence of RL online learning (influencing human physical fatigue); 5) "Action divergence" to measure offline policy optimality as in Equation (6). For all the metrics, better performance is associated with smaller/shorter outcomes.

**Questions Addressed.** Our real experimental results aim at answering the following questions:
1) Is RIIV an effective method for improving IMU sensor data quality in our data-centered solution framework?
2) Can offline normative human walking policy be further adapted and customized for individual participants via robot online learning to achieve optimal human-robot interaction?
3) Is human adaptation alone sufficient to improve performance with reduced effort?
4) Is there evidence that both human and robot co-adapted to achieve optimal interaction
5) To achieve optimal interaction between the human and the robot, what are essential control cost objectives to be considered in RL design?
6) Is the method generalizable to other locomotion tasks beyond level-ground walking?

**Q1: (Offline Benchmark Study) Our RIIV method is practically effective in capturing invariant normative walking characteristics while directly accounting for sensor and actuator noise in real environments, thereby improving offline policy optimality or action divergence.** From Figure 2 and Table 2, we can clearly see advantages of using RIIV procedure over the Direct method (Section 3.4) to process raw IMU sensor data. 1) Firstly, RIIV results in significantly lower training cost and faster convergence than the Direct Method. As illustrated in the four bar charts in Figure 2, the RIIV method (green bar) reduces the action divergence effect more greatly than the Direct Method (orange bar) does, indicating that RIIV more accurately aligns with true human walking characteristics. 2) Next, RIIV has shown to be capable of accounting for significant uncertainties inherent in physical sensing and actuation, as demonstrated by the green bar with its values closer to the ground

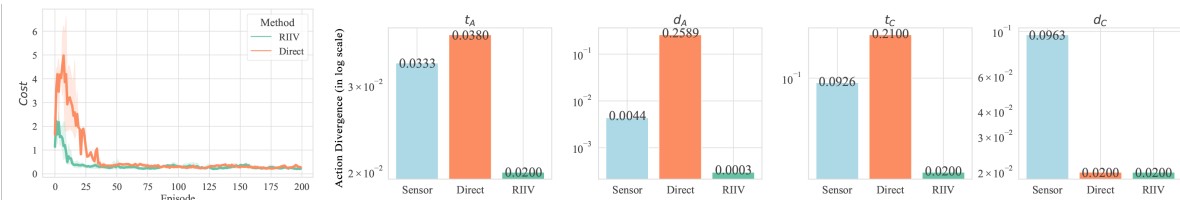

*Figure 2.* Offline learning outcomes as evidence of the essential role of processing raw sensor measurements in AIP as a data-centric method. (Left): Comparison of cost performance, Equation (7), using Direct and RIIV, the shaded regions represent the 95 % confidence range of the five random seeds. (Right 4 panels): The MoCap data is used as ground truth in the comparisons, where action divergence (AD) as in Equation (6) was measured (the closer to 0 the better.): "blue" is AD between IMU sensed data and the truth; "orange" is AD between Direct and Truth; and "green" is AD between RIIV and Truth.

*Table 1.* Performance of AIP method in terms of stage cost, peak knee error, and EMG effort.

| Performance Evaluations | Beginning of Online Training (Offline IL policy) | | | | | End of Online Training | | | | |
|---|---|---|---|---|---|---|---|---|---|---|
| Human Participant | 1 | 2 | 3 | 4 | 5 | 1 | 2 | 3 | 4 | 5 |
| Stage Cost | $0.94 \pm 0.16$ | $0.95 \pm 0.23$ | $1.1 \pm 0.47$ | $1.42 \pm 0.57$ | $0.99 \pm 0.19$ | $0.43 \pm 0.06$ | $0.49 \pm 0.12$ | $0.37 \pm 0.06$ | $0.35 \pm 0.08$ | $0.49 \pm 0.09$ |
| Training Peak Knee Error | N/A | N/A | N/A | N/A | N/A | $0.39 \pm 0.05$ | $0.23 \pm 0.04$ | $0.16 \pm 0.14$ | $0.37 \pm 0.27$ | $0.22 \pm 0.07$ |
| Training EMG Effort | N/A | N/A | N/A | N/A | N/A | $0.54 \pm 0.03$ | $0.59 \pm 0.13$ | $0.43 \pm 0.09$ | $0.57 \pm 0.14$ | $0.65 \pm 0.09$ |
| Evaluation Peak Knee Error | $0.48 \pm 0.09$ | $0.33 \pm 0.06$ | $0.28 \pm 0.05$ | $0.4 \pm 0.2$ | $1.12 \pm 0.15$ | $0.41 \pm 0.02$ | $0.23 \pm 0.04$ | $0.18 \pm 0.18$ | $0.38 \pm 0.07$ | $0.56 \pm 0.12$ |
| Evaluation EMG Effort | $0.66 \pm 0.01$ | $0.96 \pm 0.3$ | $0.55 \pm 0.09$ | $0.8 \pm 0.27$ | $0.92 \pm 0.17$ | $0.52 \pm 0.05$ | $0.32 \pm 0.02$ | $0.42 \pm 0.07$ | $0.38 \pm 0.02$ | $0.41 \pm 0.07$ |
| | Baseline without Exosuit Assistance | | | | | End of Online Training | | | | |
| EMG Effort | $0.631 \pm 0.06$ | $1.14 \pm 0.03$ | $0.68 \pm 0.04$ | $0.6 \pm 0.03$ | $0.71 \pm 0.06$ | -14.4% | -48.2% | -36.8% | -5% | -8.5% |

*Table 2.* Statistic data of Figure 2. The MoCap Video data is used as ground truth in the comparisons, where action divergence (AD) as in Equation (6) was measured (the closer to 0 the better.)

| Action Divergence | Sensor | DIRECT | RIIV |
|---|---|---|---|
| $t_A$ | $0.03 \pm 0.003$ | $0.038 \pm 0.002$ | $0.02 \pm 0.002$ |
| $d_A$ | $0.004 \pm 0.01$ | $0.259 \pm 0.015$ | $0.0003 \pm 0.007$ |
| $t_C$ | $0.0926 \pm 0.03$ | $0.21 \pm 0.025$ | $0.02 \pm 0.003$ |
| $d_C$ | $0.0963 \pm 0.02$ | $0.02 \pm 0.009$ | $0.02 \pm 0.01$ |

truth, especially for $t_A$, $t_c$, and $d_c$, where there are notable discrepancies between raw sensor data and the ground truth, and also, a rather significant delay in the actuator due to inflation/deflation time.

**Q2: Online learning effectively adapted the initial offline policy to provide personalized control for individual participants and enable robust performance in human-robot normative walking.**

From Table 1, although the offline policy enables walking, it does not achieve optimal performance in terms of cost, kinematic error, and EMG measures. As shown in Figure 3 and Table 1, while the offline policy directly benefits participants 2 and 3 in terms of reduced EMG effort (below baseline shown by dashed line), it fails to do so for participants 1, 4 and 5. Through online training, Performance metrics improve for reduced cost and kinematic error, and most importantly, reduced EMG effort for all subjects. Notice additionally that online training resulted in consistent and robust assistance to human walking. From Figure 3 and

Table 1, a significant intra-subject variance and inter-subject variance is apparent. At the initial online learning stage (gait cycle 1), the same offline policy produced varying performances across different participants. However, by the end of training, the cost consistently converged to similar values of around 0.5, which indicates that online training has effectively customized the initial offline policy for each individual, allowing all participants to reach normative walking patterns with at least a 20% reduction in EMG effort.

**Q3: Human adaptation alone cannot sufficiently improve human walking performance.** To isolate the effect of co-adaptation, a feasible approach was to disable the robot's adaptation in order to observe human adaptation. This is implemented by freezing policy updates. The results are shown in Figure 7, where in row A (baseline), participants walked naturally and showed their natural gait pattern; in row C (after online training), the policy is already optimized, so no further human adaptation is needed. In both cases, we observe no sign of human adaptation. In row B (using offline-learned, and fixed policy), this offline policy is not optimal and needs to be personalized. This is where we speculate that the participants may realize the mismatch and attempt to adapt. But due to a lack of clear or consistent direction, the participants' responses did not exhibit any discernible pattern or trend. For example, P1 shows delayed timing in $t_A$, while others do not exhibit this behavior. Similarly, P5 demonstrated a slightly earlier duration in $d_C$, which was not seen in other participants. In contrast, evidence of human-robot co-adaptation during

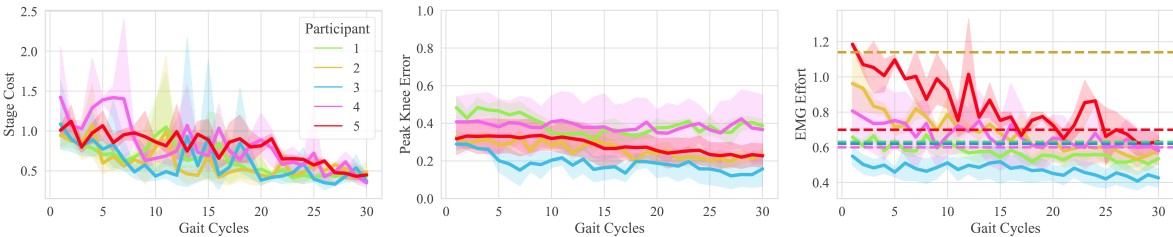

*Figure 3.* Results of online training for all five participants where the shaded regions indicate the 95% confidence interval for the three online trials. The dashed lines are respectively the baseline human walking EMG effort without exosuit assistance. Participant 1 provided the offline policy.

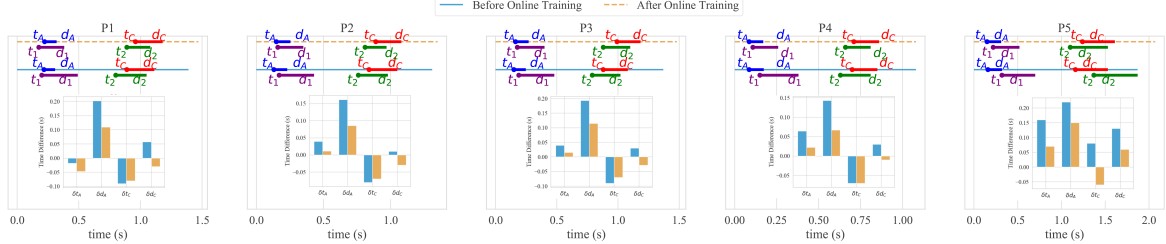

*Figure 4.* Timing and duration in state and control variables to demonstrate adaptation taking place during online learning. In the top panel above the bar charts, the blue line segment is $d_A$, the red is $d_C$, the purple is $d_1$ and the green is $d_2$. The bar plots show the mean differences in timing and duration between respective actual human walking measurements and those of the robot control. Specifialy, $\delta t_A = t_1 - t_A$, $\delta d_A = d_1 - d_A$, $\delta t_C = t_2 - t_C$, and $\delta d_C = d_2 - d_C$.

online learning was clear, the benefit of which is shown in the discussion Q4 below.

## Q4: Human and robot co-adapted to achieve normative walking with reduced human EMG Effort

1) Refer to Table 1, online training of robot control has led to normative walking, as measured by the peak knee angle approaching that during normative walking (small peak knee error), and reduced EMG effort for all participants. This is a result of online co-adaptation between the human and the robot. To see that, we show next how robot control has taken effect by looking into measurable human walking states. 2) Let's examine the duration of human stance phase ($d_A$) and swing phase ($d_C$) of offline policy and after online learning. Note that the respective duration has changed little (refer to the top row of Figure 4 above the bar charts with more details provided in (Figure 9), and Figure 8a & b). This is because the participants walk naturally and thus maintains their normative walking patterns. 3) In the meantime, note that the robot has reduced its stance duration ($d_1$) and swing duration ($d_2$) to accommodate soft actuator deployment delays (refer to the top row of Figure 4 above the bar charts). 4) Next, if we inspect the robot control onset timing $t_1$ for stance, it varied around human stance timing $t_A$ (refer to the bar plots in Figure 4 with more details shown in Figure 9 B). As the soft actuators are to provide leg support for stance, the human responses could vary depending on how

they weigh the importance of reducing effort during this less effort demanding phase of walking. 5) The swing phase soft actuator onset timing $t_2$, however, has adapted to be ahead of the human actual start of swing $t_C$. Inflating actuators in this phase is critical to reduce human effort of lifting the leg and swing it forward. Note that the initial policy from offline learning also resulted in an even earlier swing onset $t_2$, an outcome that may be caused by out-of-distribution effect as there was no soft actuator deployment during offline training. Consequently, to accommodate soft actuator delays there has to be an early onset, but cannot be too early so that the soft actuator is in the way of a normal knee swing flexion (reduce the peak knee angle error). Additional details about addressing the OOD issue can be found at Appendix G.

**Q5 (Ablation Study) Both safe regulation of joint kinematics and reduction of human EMG effort are necessary to achieve stable human-robot normative walking.** We performed an ablation study on the cost objective function. Our proposed performance index, which incorporates both EMG effort and kinematic error or state error, demonstrates superior performance. By balancing the reduction of EMG effort and adherence to state error tolerance and safety constraints, the RL controller optimizes both aspects of the user's walking behavior. Refer to Figure 3, this balanced approach leads to convergence, stability, and significant improvements in the user's mobility, as evidenced by lower stage cost, peak knee error, and decreased EMG effort.

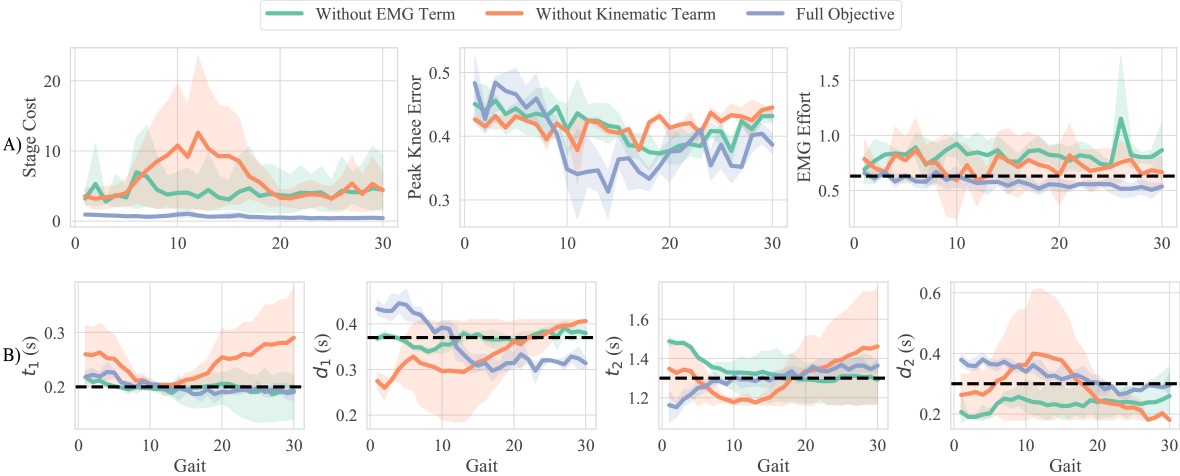

*Figure 5.* Ablation study: Learning performance under different formulation of objective functions associated with ablation study 1) Without EMG effort in the control objective function 2) Without kinematic term in the control objective function 3) Full version of objective function. **Row A**: Results of online training for three different objective functions. **Row B**: Results of robot controller timing for three different objective functions. The shaded regions represent the 95 % confidence range of the three experiment trials. The x-axis is the number of gaits. The black dash lines are the reference BASELINE from human normative walking profiles.

However, if only EMG effort or Kinematic error was used in Equation 10, not only the EMG did not reduce but also it resulted in a significant learning variance. The absence of state kinematic error in the cost function (Figure 5 **Orange line**) resulted in failure to maintain normative walking patterns, which led to increased EMG levels and overall less effective assistance . The absence of EMG effort in the performance index (Figure 5 **Green line**) leads to a lack of focus on reducing muscle activity. Consequently soft exosuit failed to provide the necessary support to reduce muscular strain, resulting in increased EMG levels .

**Q6 AIP is generalizable to incline walking.** As a first step of extension, we developed a new incline walking platform to validate the feasibility of AIP in more complex locomotion tasks. As shown in Figure 10, which are results from a 7-degree incline treadmill walking, increasing the incline significantly alters gait kinematics. However, as shown in Panel C, the AIP method remains robust, consistently reducing EMG effort below BASELINE levels across all participants.

## 5. Limitation and Future Work

Firstly, the AIP method has been tested for a single side control of one leg. Extending it to bilateral control introduces challenges in coordinating and synchronizing assistance across limbs. Multi-agent RL offers a promising direction to address this open question. Secondly, RL's reliance on extensive data is a persistent limitation. Methods such as LNSS (Zhong et al., 2023a), ATD (Zhong et al., 2023b), and other techniques such as transfer learning are promising directions

under active development, as they have been shown effective to accelerate convergence and reduce user-specific data needs. Additionally, our roadmap also includes extending the method beyond basic treadmill-simulated level-ground walking to encompass more complex and realistic tasks and scenarios, such as walking at variable speeds, incline and decline walking, among others. Another impactful future research could expand this approach to consider a broader and more diverse population, integrate complex locomotion patterns, and explore long-term adaptation.

## 6. Conclusion

In this study, we introduce the AIP method to address the challenges of soft exosuit-assisted human walking by bridging offline and online learning, rather than developing yet another RL method. By leveraging existing RL algorithms (dHDP), AIP enables personalized assistance to reduce human physical effort during normative walking. AIP improves data quality through the RIIV method, integrating offline imitation learning with RIIV and online dHDP learning for data-efficient, personalized assistance. Our framework, implemented and demonstrated in a physical environment, overcomes the challenge of optimizing human-robot interaction **without a dedicated simulator or dynamic model**. We validated our approach on five participants without selectively presenting results, demonstrating a simple yet robust solution for safe robot control. The co-adaptation between human and robot effectively mitigates actuator delays, reducing muscular effort and enhancing human-robot synergy. This work paves the way for personalized robotic assistance in rehabilitation and performance enhancement.

## Acknowledgment

Zhong and Si's research was supported in part by the National Science Foundation under Grant No. 2211740. Yumbla, Soltanian, and Zhang's research was supported in part by the National Science Foundation under Grant No. 1944833 and in part by Arizona Biomedical Research Centre under Grant No. RFGA2022-010-08. The authors thank Jahnav Rokalaboina and Nick Kirkby for their support in collecting data for the incline experiments.

## Impact Statement

This paper is the first to demonstrate the application of reinforcement learning (RL) for controlling a soft exosuit to assist human walking with reduced effort. Our study specifically focuses on providing walking assistance to healthy individuals walking on a flat treadmill. This study is not attended for walking rehabilitation applications, which will be considered in our future work. What we have shown in this study is an important first step toward future broader applications of the technology. We chose this setting because implementing such technology without thoroughly testing it on healthy participants under the most common condition (such as level ground) could raise significant societal concerns. Notably, assistive walking technologies for unimpaired and impaired populations are typically approached separately, each with distinct objectives.

In the case of unimpaired individuals, as addressed in our study, the goal is to reduce the effort required for walking while still achieving a normative gait—consistent with the aim stated in our paper's title. In contrast, wearable robots designed for individuals with physical impairments have different priorities. For these populations, improving gait symmetry, balance, and walking speed takes precedence over reducing effort. This is due, in part, to compelling evidence that irregular gait patterns, such as those seen in individuals with unilateral lower limb amputations, can lead to secondary complications if not properly addressed. Asymmetrical gait, for example, is commonly observed in amputees and is associated with secondary health issues like osteoarthritis in the non-amputated joints (Hof et al., 2007; Adamczyk & Kuo, 2014) and lower back pain (Ehde et al., 2001).

Given these considerations and the outcomes of this study, we believe that our current AIP framework has the potential to be further developed for studies focusing on assistive walking for individuals with physical impairments and have the potential to scale up such as walking at varying speed, on different surfaces, ascending and descending stairs. However, this would require redefining the control design objectives, rethinking offline data collection strategies, and developing a new experimental protocol tailored to the unique needs of this population. These adjustments are crucial to effectively address the distinct challenges of impaired gait and to ensure the success of such an extension.

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

## A. Participant Information and IRB approve

Five healthy individuals (3 male and 2 female, participated in the study under a protocol approved by the Institutional Review Board (IRB ID#: STUDY00011110 for level ground walking) and ( IRB ID#: STUDY00019873 for incline walking) The average height, weight, and age of the recruited participants were 163 ± 8 cm, 66.1 ± 11.6 kg, and 28 ± 1.9 years, respectively. The complete anthropometric data of the subjects can be seen in Table 3.

Figure 6 is a summary of histograms of the range of gait patterns which directly illustrate gait diversity of the participants. These data show that the participants' gait patterns span a broad spectrum, which covers safe and normative walking ranges as reported in well-established human dynamic walking literature. We therefore consider our experimental data capture commonly observed gait variations.

*Table 3.* Subject participants' anthropometric data.

| Subject | Gender | Age | Weight (kg) | Height (m) |
|---------|--------|-----|-------------|------------|
| S1 | M | 26 | 76 | 1.75 |
| S2 | F | 27 | 52 | 1.54 |
| S3 | M | 28 | 79 | 1.65 |
| S4 | F | 31 | 57.5 | 1.58 |
| S5 | M | 28 | 80 | 1.72 |

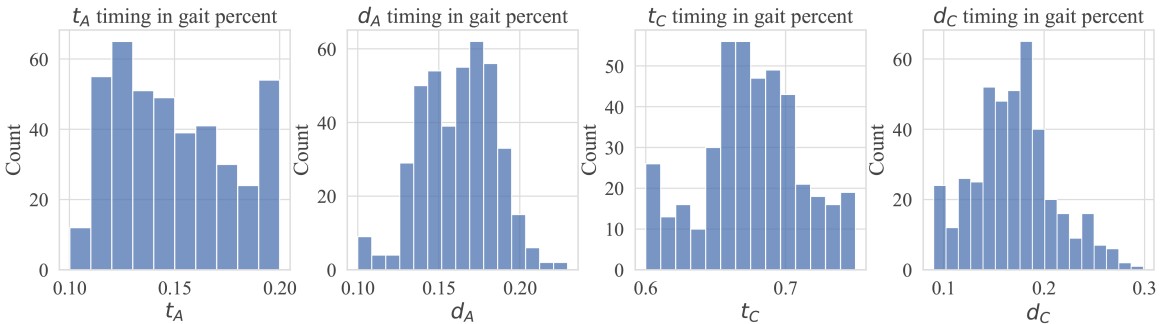

*Figure 6.* Histogram to show Diversity of Gait Patterns across all the participants. All recorded gaits show coverage of possibles region within the normative walking ranges ( $t_A \in [0.1, 0.2]$, $d_A \in [0.1, 0.3]$, $t_C \in [0.6, 0.75]$, $d_C \in [0.1, 0.3]$)

## B. Additional Related Work

**RL successes with and without simulated environments.** The most celebrated reinforcement learning (RL) achievements with superhuman performance are in playing computer games (Silver et al., 2014; Mnih et al., 2015). These successes are largely attributable to the use of unbiased simulation environments, which provide extensive and repeatable training data. However, the simulator-based successes have rarely been duplicated in the real physical world. High-fidelity simulators are often prohibitively expensive or even impossible to construct due to the complex dynamics, limitations in assessing and representing inherently uncertain physical systems, such as sensor and actuator noise, communication delays and other factors (Rao et al., 2020; Niu et al., 2022; Nikovski et al., 2024). Nonetheless, RL has been directly applied to physical systems without the use of simulators. For example, (Inoue et al., 2017) presents a method to enable industrial robots to perform high-precision assembly tasks (such as the peg-in-hole) by training an LSTM using reinforcement learning directly on the physical device. There have been some other successful demonstrations of RL agents interacting with simulated raw environments instead of simulators providing directly accessible state-action-reward data. For example, (Hilleli & El-Yaniv, 2018) trains RL agents for autonomous highway steering using raw image sequences from a simulated environment. The VPT (Baker et al., 2022) is a semi-supervised imitation learning method, where an inverse dynamics model (IDM) is trained with labeled data to generate pseudo-labels for a vast amount of unlabeled online videos. This allows for training a behavioral prior that exhibits nontrivial zero-shot capabilities and can be fine-tuned using imitation learning and reinforcement learning to perform complex tasks. The method is shown to achieve significant results in the Minecraft

Game, especially the crafting diamond tools, which were impossible for RL alone previously. The AR2-D2 (Duan et al., 2023) allows users to record themselves manipulating objects, and the data is then used to train a real robot to perform similar tasks. Despite these efforts, these studies still depend on gathering large amounts of data, often requiring hundreds of hours of long sequences of video or episodes, for RL to effectively converge. There is a **significant gap** in research addressing RL applications that operate under limited data conditions, particularly those with fewer than a few hundred state transitions. Another compounding factor that significantly complicates the problem in data-scarce environments is the human-in-the-loop effect, which is difficult to model or build a simulator for the human-robot interacting dynamics. These issues remain largely unexplored.

**Offline to Online RL** Offline-to-online RL has been explored through various approaches. (Zhang et al., 2023) introduce policy regularization via expansion and implement a Boltzmann action selection strategy. Several studies, including (Lee et al., 2022), (Zhao et al., 2023a), and (Zhao et al., 2023b), utilize an ensemble of pessimistic value functions to mitigate distributional shift, which serves as an implicit form of conservatism. (Nakamoto et al., 2024) propose training an additional value function to address over-conservatism issues caused by the initialized value function in the offline phase. (Niu et al., 2022) develop a dynamics-aware policy evaluation scheme to bridge the dynamic gap between source and target domains. (Li et al., 2023) introduce a policy regularization term for trust-region-style updates. (Lei et al., 2023) use an on-policy optimization method that unifies both offline and online training without extra regularization. However nearly all of the current offline to online RL collects the offline data based on extensive simulations with millions of samples and none of them have demonstrated directly learning from physical environment using LIMITED data for an offline policy. Our AIP approach is one of the first realization that does not require simulator to provide offline data and We developed our RIIV-based, data centric approach not only reduces intra-person step length variations but also reduces inter-person variation. Because of our innovative data-centric approach, plus our use of well-established methods (BC and dHDP), our AIP approach demonstrated its impressive performance as we have reported.

**dHDP algorithm.** The dHDP is one of the earliest deterministic policy gradient methods (Si & Wang, 2001). This class of algorithms is considered more effective than popular methods such as PPO (Schulman et al., 2017) and SAC (Haarnoja et al., 2018) for real-time continuous control applications. The dHDP is based on a fundamental principle that the actor adjusts the policy in the direction of the action-value gradient, and the critic updates the action-value function. This idea is considered a predecessor in a better known algorithm, the NFQCA (Hafner & Riedmiller, 2011), which in turn, is considered a predecessor of the even better known DPG algorithm (Silver et al., 2014), which, as we know, has been further developed into good algorithms, such as DDPG and SAC. In a nutshell, the dHDP is a bare-bone policy gradient method. All these methods can be made more stable by integrating experience replay and target networks, which were first introduced in the deep Q-networks (DQN) (Mnih, 2013). A more detailed discussion on dHDP and its performance evaluation in comparison to DDPG, as well as its several significant applications in complex and realistic engineering systems, such as Apache helicopter stabilization, tracking, and reconfiguration control (Enns & Si, 2003), power grid (Sun et al., 2012), chemical processes (Yang et al., 2021), fuzzy control system (Gao & Liu, 2016), Single-Axis Servo Mechanism System (El-Sousy et al., 2024), and more can be found in (Wu et al., 2024).

PPO and SAC are widely used in robotic learning based on extensive simulations or under structured environment. Both algorithms have shown great ability to learn stochastic policies by parameterizing a probability distribution over actions. These distribution-based policies facilitate exploration by sampling actions during training (Haarnoja et al., 2018; Schulman et al., 2017). However, applying PPO and SAC in exosuit robot that does not have a simulator or a dynamic model of human-robot interaction, presents true challenges.

In wearable robotics, particularly with humans in the loop, safety and comfort are of paramount importance. Human participants become frustrated if the robot operates erratically. An analogy is how people feel when they learn to ski. Falling to the ground in arbitrary ways make them feel frustrated, a mental state that adversely affect their natural behavior or even cause injuries. A stochastic policy can become a source of causing discomfort to human users. In other real time control applications such as flight control, we cannot afford erratic control actions either due to safety concerns. Perhaps because of this, to the best of our knowledge, all existing applications of PPO and SAC rely on simulators, which enable unlimited data collection, and controls are conducted in structured environments. The reliance on huge amount of data and the needs of safety assurance have significantly narrowed the pool of potentially feasible methods.

However dHDP has been shown convergent theoretically and demonstrated convergence under limited data conditions empirically in several wearable robotics applications (Wen et al., 2016a;b; 2017a; 2019; Wu et al., 2022). Additionally, dHDP has been shown its compatible performance to DDPG in many DMC robotic control problems (Wu et al., 2024).

These nice theoretical properties and practical successful studies have given us the confidence to deploy dHDP in our current study, which directly learn the exosuit control solution with 5-dim state space and 4-dim action space.

In summary, given the track record of dHDP in human-robot locomotion control applications, and also to build initial trust in human participants for this new soft robot control problem, we took the first step of using dHDP to learn in real time for soft robot assisted human walking with reduced effort. This has helped us overcome the critical issue to not rely on a mathematical model or simulation model, which are nearly impossible to obtain (Brenneis et al., 2021). While this approach has met our expectation in the current study, we recognize the limitations or potential challenges as summarized in the Limitations section, and we plans to systematically explore and evaluate different solution architectures and algorithms in future studies.

**Modeling of soft exosuit** A pertinent example of these challenges can be seen in the develo±ent and application of soft inflatable exosuits. On one hand, modeling a simulator for these devices is highly complex (Polygerinos et al., 2015). This is mainly due to the compliant nature of soft robots, which introduce physical properties that are difficult to model. For instance, several studies have shown that even obtaining a quasi-static model of torque for a soft actuator is not trivial (Nesler et al., 2018; O'Neill et al., 2022). On the other hand, the dynamic interaction between the human wearer and the robot creates a highly coupled and complex system, making it even more challenging to model accurately (Zhu et al., 2022).

**Control of rigid lower limb exoskeleton and powered prosthesis.** Rigid robotic lower-limb exoskeletons and prostheses and their controls are being actively researched or even commercially available (Huang et al., 2021; Siviy et al., 2023; Shi et al., 2019). A typical **control strategy** of these devices often focus on mimicking the kinematics of biological joints via position control (Bortole et al., 2015; Long et al., 2017). However, another control strategy, referred to as finite state machine impedance control, is often preferred especially for consideration of achieving compliant lower limb behaviors. This stratagy provides safe human-exoskeleton interactions, as biological systems are capable of in order to adapt to various environments (Azocar et al., 2020). Unfortunately, neither of the above two strategies are applicable to soft exosuit. For **control actuation** in either control strategy, rigid device control torques are generated by mechanical joint motor actuators, which can be directly used as a control parameter. In soft inflatables, however, the torque is generated from both human and the exosuit. Differentiating the two sources is difficult or nearly impossible. The natural control parameters for exosuit instead is the timing of inflation and deflation, which introduces additional delays to actuation and thus reduced stability margins have to be considered in the control design.

**Personalized controls** for individual users are common to all wearable lower limb devices, and human-in-the-loop (HIL) optimization represents several important design approaches (Koller et al., 2016; Zhang et al., 2017; Ding et al., 2018; Kim et al., 2019a; Bryan et al., 2021). They are used in open or closed-loop force or torque control to operate individual joints. In those applications, Bayesian optimization plays a key role to provide optimal controls. These methods search for an extremum on the system response surface to determine the optimal control parameters. Although these methods can customize the control strategies or parameters, they are time-consuming and lack of adaptation. A small change of the wearer requires a re-design of the control (Tu et al., 2021). It is noted that a large class of wearable rigid device controls aims at achieving reducing **metabolic cost** reflected by oxygen intake. However, it usually takes long walking time to be able to extract reliable measurements. This prohibits online and real time requirement that is highly desired for wearable robot applications. To overcome the limitations due to control strategy or optimizaiton method, data-driven **reinforcement learning** (RL)-based optimal adaptive control methods have been developed and successfully demonstrated for robot control of exoskeletons and prostheses. (Wen et al., 2019; 2017b; Li et al., 2021). In these applications, robot control relies on optimal cumulative cost/reward related to producing normative walking using directly measurable human-robot walking variables. However, these methods have been principally implemented in rigid robot devices (Huang et al., 2021), not soft inflatable exosuits, the control problem formulation is different and the solution presents unique challenges due to discussions in the above.

## C. Safety Constraints

The problem under investigation requires human physical safety and control system stability of the human-robot system. In this study, physical safety refers to that the human participants do not fall or endure injury as a result of robot control. This is ensured by imposing safety bounds to limit the soft suit inflation and inflating duration timing. The control system stability is in the same classical control sense, and we set one of the RL design objective in Equation 11 that the state regulation errors approach 0 (or practically error tolerance bounded). Our assurance of stability and safety is embedded in learning quantitatively, and guaranteed by analysis qualitatively. Our systematic data have shown that these constraints are met and

objectives achieved.

To ensure human participants walk continuously and safely, we consider several safety constrains: 1) the soft actuator pressure is limited to 206.8 kpa; 2) the control timings and inflation/deflation duration are constrained by taking reference of those during participant's normative walking as shown in Table 4, which are within realistic ranges (Zhang et al., 2020b). These physical constraints help prevent significant misalignment between controller timing and the respective gait phase during human walking. Without these constraints, it may trigger soft actuator deployment and cause discomfort to the user; and 3) the online training objective is set for the control timings to approach those during normative walking, and thus in a safe state. Equation (11) renders such state constraint where the target state variables $\tilde{s} = [\tilde{t}_A, \tilde{d}_A, \tilde{t}_C, \tilde{d}_C, \tilde{\theta}]$ (Table 4) are obtained from normative walking profile without soft actuator deployment to assist human walking:

Table 4. Safety constraint for the control timing and target state for control regulation to reach.

| Control | $t_1$ | $d_1$ | $t_2$ | $d_2$ | Target State | $\tilde{t}_A$ | $\tilde{d}_A$ | $\tilde{t}_C$ | $\tilde{d}_C$ | $\tilde{\theta}$ |
|---|---|---|---|---|---|---|---|---|---|---|
| Safety constraints (% gait phase) | [0,20] | [0,20] | [60,80] | [0,20] | normal walking (% gait phase) | 14% | 15% | 68% | 15% | $60^o$ |

$$\epsilon_s = (s - \tilde{s})^2, \tag{11}$$

where the respective error tolerance for each state variable is as shown in Table 5. They represent realistic sensing and actuation errors inherent in physical systems, and they are physically meaningful, human physiologically realistic, and validated in studies of human biomechanics such as (Zhang et al., 2020b).

Table 5. Ranges of state error tolerance that are used in learning for achieving normative walking.

| Error Tolerance | $t_A - \tilde{t}_A$ | $d_B - \tilde{d}_B$ | $t_C - \tilde{t}_C$ | $d_C - \tilde{d}_C$ | $\theta - \tilde{\theta}$ |
|---|---|---|---|---|---|
| Tolerance Range | [-5%,5%] | [-5%,5%] | [-5%,5%] | [-5%,5%] | [0, 40 deg] |

Table 6. Tolerance values that ensure human normative walking.

| State | $t_A$ | $d_B$ | $t_C$ | $d_C$ | $\theta$ |
|---|---|---|---|---|---|
| inf value | 10 % | 10% | 60% | 10% | $53^o$ |
| sup value | 20 % | 30% | 75% | 30% | $78^o$ |

## D. Hardware Details

**Soft inflatable actuators** are designed to generate extension torque, fabricated from nylon fabric and thermoplastic polyurethane to ensure a transparent interaction with the user. When the knee is flexed and the actuators are inflated, they apply extension torque to the knee joint. These actuators are strategically positioned in the popliteal fossa to aid in knee extension. Further design details of the soft inflatable exosuit can be found in (Sridar et al., 2018; Sridar et al., 2020).

The electro-pneumatic system that controls the real-time inflation of the exosuit includes:
**a microcontroller (Raspberry Pi),**
**solenoid valves (MHE3-MS1H, Festo, Hauppauge, NY)** for switching between inflation and deflation,
**pressure sensors (ASDXAVX100PGAA5, Honeywell International Inc., Morris Plains, NJ)** for monitoring the internal pressure of the actuators.

Kinematic data were collected using:
**a camera-based motion capture system (T40s, VICON Inc., Los Angeles, CA)** sampling at 100 Hz.
**EMG sensors (Delsys Trigno, Delsys, Natick, MA)** were used to capture muscle activity, sampled at 2000 Hz.
**An instrumented treadmill (Bertec Inc., Columbus, OH)** was used as the platform for the walking trials. The treadmill is equipped with force plates that measure the user's ground reaction forces at a sample rate of 2000 kHz.
**An IMU motion capture system (Ultium Motion, Noraxon Inc., Scottsdale, AZ)** was used to detect the maximum knee joint angle.

EMG sensors were placed on both legs over three muscles of interest: *vastus lateralis* (VL), *biceps femoris* (BF) and *rectus femoris* (RF). The raw EMG data were first band-pass filtered (Butterworth, 4th order, 20 Hz and 450 Hz cutoff frequencies). The profile of the signal was obtained by computing the root-mean-square envelope using a moving window of 250 ms. The integral of the envelope was computed for each gait cycle to quantify the overall muscular effort.

## E. Experimentation, Hyperparameters and Implementation Details

We use PyTorch for all implementations. All results were obtained using our desktop with Intel Core i9-12900K processor.**Experimentation.** The experiments consisted of two sets of walking: an offline, normative walking session with the exosuit attached but not inflated, and an online walking session with RL-controlled exosuit inflation and deflation. Data of one participant was recorded during offline walking for one experiment session. The offline session lasted around 10 minutes of about 170 steps. For the online human-robot collaborative walking, three sessions were performed for each participant. Each online session began with the controller initialized to the learned offline policy from IL, and lasted 10 minutes of about 150 steps. All participants walked at a constant speed of 1 m/s on the treadmill during all experimental sessions, **Safety constraints** were imposed as discussed in Section 3.3. **Data Collection for Offline Training.** After a two-minute warm-up period for participant 1 to get accustomed to the experimental setup and walking speed, data collection began. MoCap video data of the state variables $s$ in Equation (1) were collected. MoCap data were synchronized with the time sequences of the state variables to provide control target for offline training. **Data Collection for Online Training.** After a two minute warm up period, the learned offline policy (refer to Section 3.4) was used for initializing the online RL controller for all participants. To mitigate environment noise and intra-human variance, a consecutive 5 steps were used to obtain one gait sample, resulting in a total of 30 gait samples. The RL policy update was performed for every gait sample. **Performance Evaluations.** Evaluation sessions were performed after online learning convergence upon meeting criteria (Table 5). Each participant rested for 5 minutes after online learning prior to evaluation which involves walking of 100 steps in about 6 minutes. Evaluation data were processed using similar procedures to those used in processing online learning data.

### E.1. Offline Training Procedure

The offline training consist with 200 episode. An Episode start with the first offline data in the dataset $\overline{\mathcal{D}}$ to the end of the dataset with total data points of 150. For each training trial, we use an off-policy exploration strategy, adding Gaussian noise $\mathcal{N}(0, 0.05)$ to each control. The algorithm hyperparameter for offline training is as Table 7.

| Hyperparameter | Value |
|---|---|
| Exploration noise | $\mathcal{N}(0, 0.05)$ |
| Noise clip | $\pm\, 0.5$ |
| Policy update frequency | 2 |
| Batch size | 32 |
| Buffer size | 200 |
| $\gamma$ | 0.95 |
| $\tau$ | 0.1 |
| Adam Learning rate | 0.001 |

*Table 7.* Hyper Parameters used for offline training

### E.2. Online Training Procedure

For the online human-robot collaborative walking, three sessions were performed for each participant. Each online session began with the controller initialized to the learned offline policy from IL, and lasted 10 minutes of about 150 steps. All participants walked at a constant speed of 1 m/s on the treadmill during all experimental sessions, Safety constraints were imposed as discussed in Section 3.3. The algorithm hyperparameter for offline training is as Table 8.

### E.3. Network Structure and optimizer

The actor-critic networks in DHDP are implemented by feedforward neural networks with two layers of weights. Each layer has 256 hidden nodes with rectified linear units (ReLU) for both the actor and critic. The input layer of actor has the same dimension as observation state. The output layer of the actor has the same dimension as action requirement with a tanh unit.

| Hyperparameter | Value |
|---|---|
| Exploration noise | $\mathcal{N}(0, 0.01)$ |
| Noise clip | $\pm 0.1$ |
| Policy update frequency | 2 |
| Batch size | 5 |
| Buffer size | 20 |
| $\gamma$ | 0.95 |
| $\tau$ | 0.4 |
| Adam Learning rate | 0.001 |

*Table 8.* Hyper Parameters used for online training

Critic receives both state and action as input to THE first layer and the output layer of critic has 1 linear unit to produce $Q$ value. Network parameters are updated using Adam optimizer with a learning rate of $10^{-3}$.

### E.4. Code

For Code and data, please visit https://github.com/JennieSi-Lab-RLOC/ICML2025-AIP

## F. dHDP solution and properties

To find thd dHDP solution, let the critic value as Equation 8 be $Q_\theta$ where $\theta$ denotes the critic weights that are to be learned by using dHDP. Specifically, weight updates were performed to minimize the loss as a function of the weights ($\theta$):

$$L(\theta) = \mathbb{E}_{s \sim p_\pi, u \sim \pi} \left[ (y - Q_\theta(s_k, u_k))^2 \right], \tag{12}$$

where in the above, $y$ denotes the critic target. Accordingly, the actor weights (denoted by ($\phi$)) are updated by applying the chain rule to the total return from the start distribution $J$ with respect to the policy parameter ($\phi$):

$$\nabla_\phi J(\phi) = \mathbb{E}_{s \sim p_{\pi_\phi}} \left[ \nabla_u Q_\theta(s_k, u_k)|_{u_k = \pi_\phi(s_k)} \nabla_\phi \pi_\phi(s_k) \right]. \tag{13}$$

The update rules for the critic and the actor, respectively are:

$$\begin{aligned} \theta &\leftarrow \theta + \alpha \nabla_\theta L(\theta), \\ \phi &\leftarrow \phi + \alpha \nabla_\phi J(\phi), \end{aligned} \tag{14}$$

where $\alpha$ is the learning rate.

Here we analyze and characterize properties of the learning process and the control performance, specifically those related to learning convergence, solution optimality, and stability as a result of online dHDP learning. In the following, we express the exosuit control system with the following general nonlinear dynamics for the ease of discussion although this model is unknow, and our offline to online learning is completely data-driven.

$$s_{k+1} = f(s_k, u_k), k = 0, 1, ... \tag{15}$$

where $s \in \mathbb{R}^5$ and $u \in \mathbb{R}^4$ are defined in Equations 1, 2, respectively, $k$ denotes discrete time steps.

The objective of optimal control is to find a control policy that can stabilize system (15) and minimize the cost-to-go in Equation (8).

According to the Bellman optimality principle, the optimal cost-to-go satisfies the following relationship,

$$Q^*(s_k, u_k) = c_k(s_k, u_k) + \gamma Q^*(s_{k+1}, \pi^*(s_{k+1})), \tag{16}$$

and the optimal control law $\pi^*$ can be expressed as

$$\pi^*(s_k) = arg \min_{u_k} Q^*(s_k, u_k), \tag{17}$$

where $Q^*(s_k, u_k)$ is the state-action value function corresponding to the optimal control policy $\pi^*(s_k)$.

We need the following definition and assumption to develop our results.

**Definition 1.** (Stabilizable System) A nonlinear dynamical system is said to be stabilizable on a compact set $\Omega \in \mathbb{R}^n$, if for all initial states $s_0 \in \Omega$, there exists a control sequence $u_0, u_1, \ldots, u_k, \ldots$ such that the state $s_k \to s^e$ as $k \to \infty$ where $s^e$ is a equilibrium point.

**Assumption F.1.** System (15) is controllable and stabilizable. The system state $s_k = s^e$ is an equilibrium of the system under the control $u_k = \pi(s_k) = u^e$ for $s_k = s^e$, i.e., $f(s^e, u^e) = s^e$. The feedback control sequence $u_k$ is determined from control policy $\pi$ represented by the actor neural network, and in the most general case is bounded by actuator saturation.

**Assumption F.2.** The stage cost function $c_k(s_k, u_k)$ is finite, continuous in $s_k$ and $u_k$, and positive semi-definite with $c_k(s_k, u_k) = 0$ if and only if $s_k = s^e$ and $u_k = u^e$.

Note that the above assumptions are reasonable and realistic, as they are under the presumptions that a person who can use a exosuit to assist walking can reach an equilibrium state that they can achieve normative walking while their muscle activities are reduced to a level less than that without wearing assistance.

As a actor-critic method, dHDP solve the Bellman's optimality by learning to approximate both policy and value functions where actor refers to the learned policy and critic refers to the learned value. An actor-critic algorithm starts with an initial value, e.g., $Q_0(s, u) = 0$ and an initial arbitrary policy $\pi_0$. Then for $i = 0, 1, 2, \ldots$, it iterates between policy update and policy evaluation steps.

$$Q_{i+1}(s_k, u_k) = c_k(s_k, u_k) + \gamma Q_i(s_{k+1}, \pi_i(s_{k+1})), \tag{18}$$

and

$$\pi_i(s_k) = \arg\min_{u_k} Q_i(s_k, u_k). \tag{19}$$

Or by combining (18) and (19), we have

$$Q_{i+1}(s_k, u_k) = c_k(s_k, u_k) + \gamma \min_{u_{k+1}} Q_i(s_{k+1}, u_{k+1}). \tag{20}$$

**Theorem F.3.** *Let Assumptions F.1 and F.2 hold. Let $Q_i$ be the sequence of estimated Q values starting from $Q_0 = 0$ at $i$th update of RL agent. For policy $\pi_i$, its actor network weights are updated based on the policy gradient estimator (14), and the controls are bounded by the output function of the action network. Then*

*(1) **Bounded:** there is an upper bound $Y$ such that $0 \le Q_i(s_k, u_k) \le Y$, for $i = 1, 2, \ldots$.*

*(2) $Q_i$ is a non-decreasing sequence satisfying $Q_i(s_k, u_k) \le Q_{i+1}(s_k, u_k), \forall i$.*

*(3) **Convergence:** the limit of the sequence, $Q_\infty(s_k, u_k) = \lim_{i \to \infty} Q_i(s_k, u_k)$, satisfies*

$$Q_\infty(s_k, u_k) = c_k(s_k, u_k) + \gamma \min_{u_{k+1}} Q_\infty(s_{k+1}, u_{k+1}). \tag{21}$$

*(4) **Optimality:** the Q-value sequence $Q_i(s_k, u_k)$ and the corresponding policy $\pi_i(s_k)$, with $\pi_\infty(s_k) = \lim_{i \to \infty} \pi_i(s_k)$, converge to the optimal value $Q^*$ and optimal policy $\pi^*$, respectively:*

$$\pi_\infty(s_k) = \pi^*(s_k), \tag{22}$$

$$Q_\infty(s_k, u_k) = Q^*(s_k, u_k). \tag{23}$$

*Proof*. (1) Let $\eta(s_k)$ be a deterministic control policy represented by a neural network which is a continuous mapping from $s_k$ in stochastic environment $E$. Let $Z_0(\cdot) = 0$, and $Z_i$ be updated by

$$Z_{i+1}(s_k, u_k) = c_k(s_k, u_k) + \gamma Z_i(s_{k+1}, \eta(s_{k+1})), \tag{24}$$

Thus, $Z_1(s_k, u_k) = c_k(s_k, u_k)$.

According to Lemma 2 in (Gao et al., 2024), we obtain

$$Z_{i+1}\left(s_k, u_k\right)$$
$$= \sum_{j=0}^{i} \gamma^j c_k\left(s_{k+j}, \eta\left(s_{k+j}\right)\right) \leq \sum_{j=0}^{\infty} \gamma^j c_k\left(s_{k+j}, \eta\left(s_{k+j}\right)\right). \tag{25}$$

If Assumption F.1 holds, $c_k(s_{k+j}, u_{k+j})$ is bounded, there exists an upper bound $Y$ such that

$$\sum_{j=0}^{\infty} \gamma^j c_k\left(s_{k+j}, \eta\left(s_{k+j}\right)\right) \leq Y, \tag{26}$$

According to Lemma 1 in (Gao et al., 2024), as $Q_{i+1}$ is the result of minimizing the right-hand side of (20), we have

$$Q_{i+1}\left(s_k, u_k\right) \leq Z_{i+1}\left(s_k, u_k\right) \leq Y, \forall i. \tag{27}$$

(2) Define a value sequence $\Phi_i$ as

$$\Phi_{i+1}\left(s_k, u_k\right) = c_k\left(s_k, u_k\right) + \gamma \Phi_i\left(s_{k+1}, \pi_{i+1}\left(s_{k+1}\right)\right), \tag{28}$$

and $\Phi_0 = Q_0 = 0$. In the following, a shorthand notation is used for $\Phi_i(s_{k+1}, \pi_{i+1}) = \Phi_i(s_{k+1}, \pi_{i+1}(s_{k+1}))$.

Since $\Phi_0\left(s_k, u_k\right) = 0$ and $Q_1\left(s_k, u_k\right) = c_k(s_k, u_k)$, and $c_k$ is positive semi-definite under Assumption F.2,

$$\Phi_0\left(s_k, u_k\right) \leq Q_1\left(s_k, u_k\right). \tag{29}$$

From (18) and (28), we get

$$Q_{i+1}\left(s_k, u_k\right) - \Phi_i\left(s_k, u_k\right)$$
$$= \gamma\left[Q_i\left(s_{k+1}, \pi_i\right) - \Phi_{i-1}\left(s_{k+1}, \pi_i\right)\right] \geq 0. \tag{30}$$

Therefore,

$$\Phi_i\left(s_k, u_k\right) \leq Q_{i+1}\left(s_k, u_k\right). \tag{31}$$

Further by using Lemma 1 in (Gao et al., 2024)

$$Q_i\left(s_k, u_k\right) \leq \Phi_i\left(s_k, u_k\right) \leq Q_{i+1}\left(s_k, u_k\right). \tag{32}$$

This completes the proof of Theorem F.3 (2).

(3) From parts (1) and (2) in the above, $Q_i$ is a monotonically non-decreasing sequence with an upper bound. Therefore, its limit exists. Let the limit be $\lim_{i \to \infty} Q_i\left(s_k, u_k\right) = Q_\infty\left(s_k, u_k\right)$.

Given $i$ and for any $u_{k+1}$, according to (18), there is

$$Q_i\left(s_k, u_k\right) \leq c_k\left(s_k, u_k\right) + \gamma Q_{i-1}\left(s_{k+1}, u_{k+1}\right). \tag{33}$$

As $Q_i$ is monotonically non-decreasing, we have

$$Q_{i-1}\left(s_k, u_k\right) \leq Q_\infty\left(s_k, u_k\right), \tag{34}$$

the following then holds

$$Q_i\left(s_k, u_k\right) \leq c_k\left(s_k, u_k\right) + \gamma \min_{u_{k+1}} Q_\infty\left(s_{k+1}, u_{k+1}\right). \tag{35}$$

As $i \to \infty$, we have

$$Q_\infty\left(s_k, u_k\right) \leq c_k\left(s_k, u_k\right) + \gamma \min_{u_{k+1}} Q_\infty\left(s_{k+1}, u_{k+1}\right). \tag{36}$$

On the other hand, since the cost-to-go function sequence satisfies

$$Q_{i+1}\left(s_k, u_k\right) = c_k\left(s_k, u_k\right) + \gamma \min_{u_{k+1}} Q_i\left(s_{k+1}, u_{k+1}\right), \tag{37}$$

applying inequality (34) as $i \to \infty$,

$$Q_\infty (s_k, u_k) \geq c_k (s_k, u_k) + \gamma \min_{u_{k+1}} Q_\infty (s_{k+1}, u_{k+1}) . \tag{38}$$

Based on (36) and (38), (21) is true. This completes the proof of Theorem F.3 (3).

(4) According to Theorem F.3 (3) and by using Equations (18) and (19), we have

$$
\begin{aligned}
Q_\infty (s_k, u_k) &= c_k (s_k, u_k) + \gamma \min_{u_{k+1}} Q_\infty (s_{k+1}, u_{k+1}) \\
&= c_k (s_k, u_k) + \gamma Q_\infty (s_{k+1}, \pi_\infty (s_{k+1})) ,
\end{aligned}
\tag{39}
$$

and

$$\pi_\infty (s_k) = \arg \min_{u_k} Q_\infty (s_k, u_k) . \tag{40}$$

Observing (39) and (40), and then (16) and (17), we can find that (22) and (23) are true. This completes the proof of Theorem F.3 (4).

**Theorem F.4.** *Let Assumptions F.1 and F.2 hold, and $Q_i$ be the sequence of estimated Q values starting from $Q_0 = 0$. For policy $\pi_i$, its actor network weights are updated based on the policy gradient estimator (14). If $Q_i$ converges to $Q_\infty$ as $\pi_i \to \pi_\infty$, then $\pi_\infty$ is a stabilizing policy.*

Proof. If Assumption F.1 holds, let $\mu(s_k)$ be a stabilizing control policy, and let its cost-to-go $\Lambda_i$ be updated by the following equation from $\Lambda_0(\cdot) = 0$,

$$\Lambda_{i+1} (s_k, u_k) = c_k (s_k, u_k) + \gamma \Lambda_i (s_{k+1}, \mu (s_{k+1})) , \tag{41}$$

We have

$$\Lambda_i (s_k, u_k) = \sum_{j=0}^{i} \gamma^j R (s_{k+j}, \mu (s_{k+j})) , \tag{42}$$

Because $\mu(s_k)$ is a stabilizing policy, if Assumption F.1 and F.2 holds, we have $s_k \to s^e$ and $c_k(s_k, u_k) \to 0$ as $k \to \infty$. Therefore, $\Lambda_i(s_k, u_k) \to 0$ as $k \to \infty$.

Next, from Lemma 1 in (Gao et al., 2024), $\pi_i$ minimizes $Q_i$, we have

$$Q_i (s_k, u_k) \leq \Lambda_i (s_k, u_k) . \tag{43}$$

Since $\Lambda_i(s_k, u_k) \to 0$ as $k \to \infty$, we have $Q_i(s_k, u_k) \to 0$ as $k \to \infty$.

From Theorem F.3 (3), we obtain $c_k(s_k, u_k) = 0$ as $k \to \infty$. Further, under Assumption F.2, $c_k(s_k, u_k) = 0$ if and only if $s_k = s^e$, we have $s_k \to s^e$ as $k \to \infty$. This completes the proof.

# G. Addressing OOD Problem

A primary challenge during online training was caused by a significant actuator delay associated with soft actuator inflation and deflation. Specifically, there is an approximately 0.2-second delay to fully inflate and 0.25-second delay to deflate. These delays could not be adequately captured during the offline imitation learning phase as offline policy was obtained without exosuit control. As shown in Figure 4, to compensate for the inflation delay, the control variable $t_1$ was significantly shifted to an earlier onset, allowing the system to anticipate the slower actuator response time. Similarly, to mitigate the impact of deflation delays, the duration variables $d_1$ and $d_2$ were substantially shortened, ensuring that the system could maintain synchronization with the human walking pattern. These adjustments were critical in aligning the actuator responses with the real-time dynamics of human movement, thereby enhancing the overall effectiveness of AIP by achieving normative walking with reduced effort while all safety constraints are met.

# H. Human Adaptation

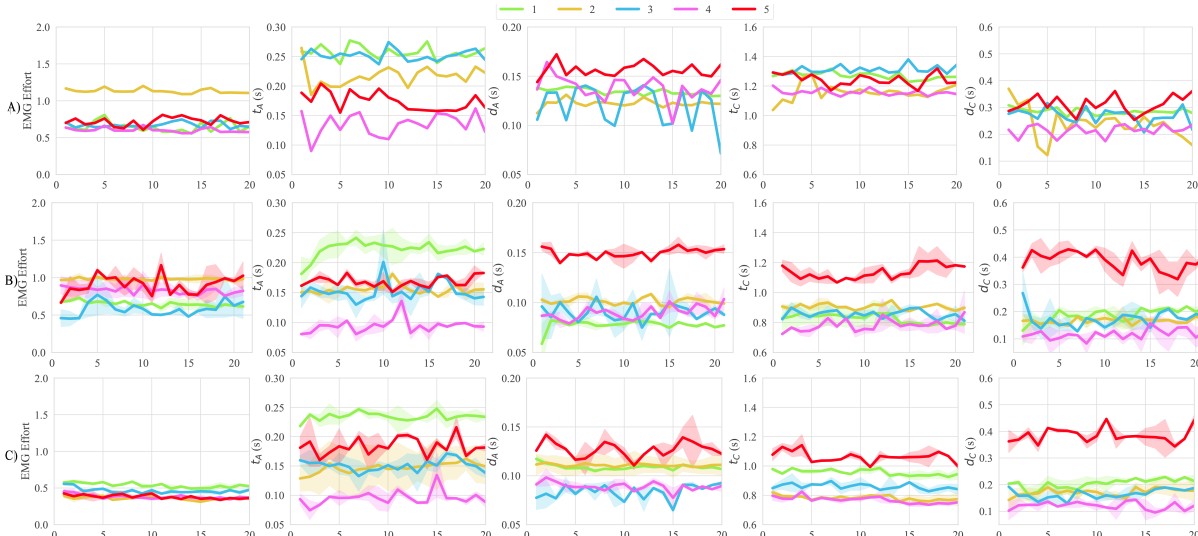

*Figure 7.* EMG profile and Characteristic timings and durations of gait trials A) **Baseline**: **without** actuating the exosuit. B) Under the **offline** policy obtained from offline training. C) Under a **trained online** policy obtained after online learning convergence. The results in row B) illustrate how participants responded and adapted to an offline-trained policy. However, their adaptations show no clear or consistent trend toward reducing EMG effort. In contrast, the human-robot co-adaptation process during online learning led to a consistent reduction in EMG effort, as demonstrated in Figures 3 and 4. The shaded regions represent the 95 % confidence range of the three experiment trials. The x-axis is the number of gaits. The results in row A) included one trial since there is no control assistance.

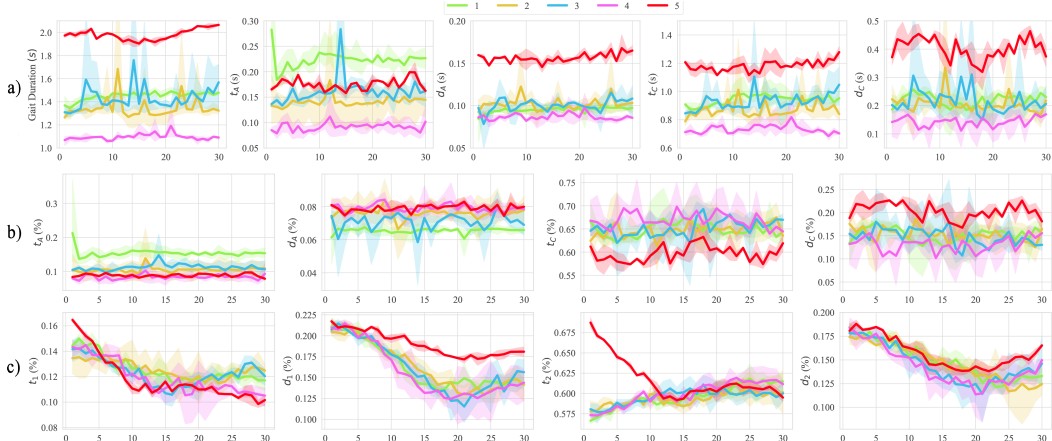

*Figure 8.* Characteristic timings and durations of gait trials: a) Raw gait data in seconds during training. b) Respective RIIV-processed data. c) RL policy during training. The shaded regions represent the 95 % confidence range of the three experiment trials. The x-axis is the number of gaits.

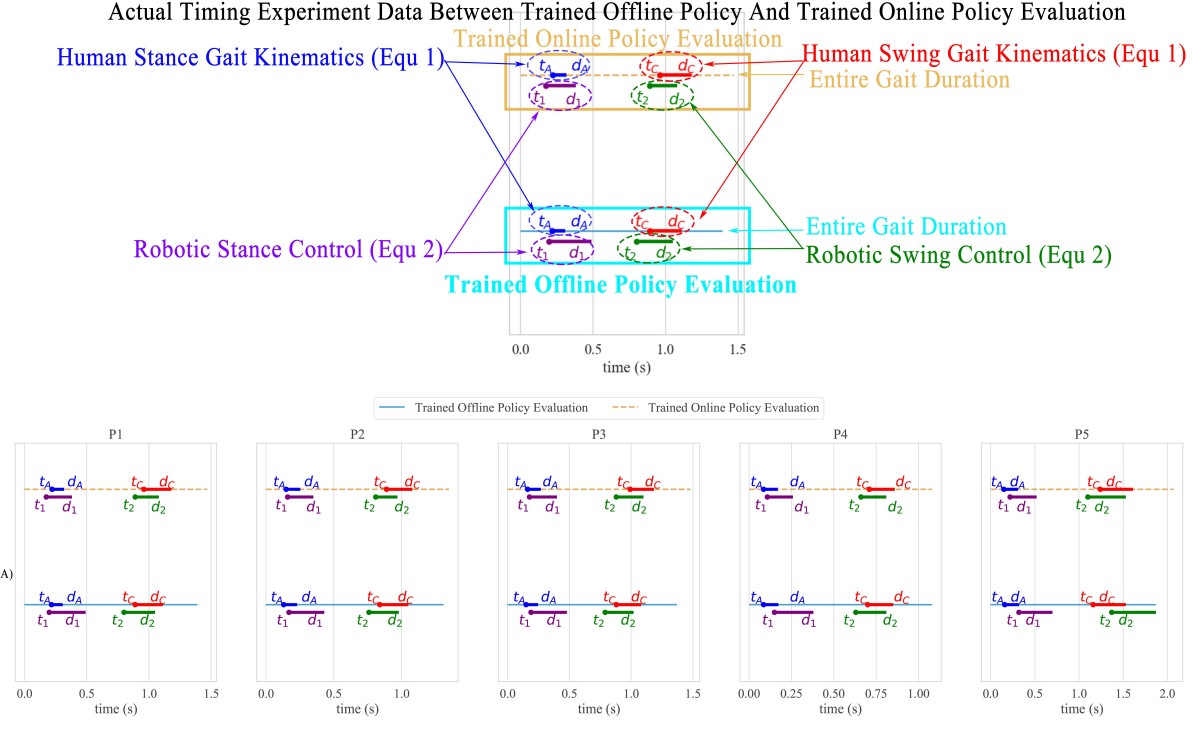

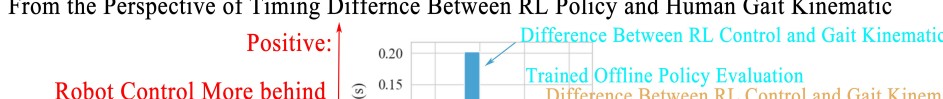

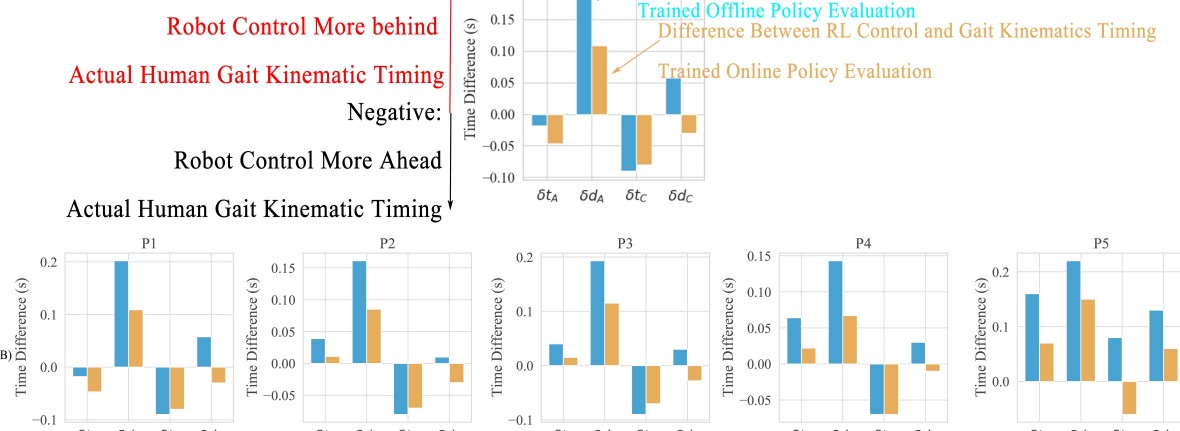

*Figure 9.* **Row A** represents actual experimental data from evaluations of both the offline-trained policy and the online-trained policy. **Row B:** explores human-robot co-adaptation by analyzing the timing differences between the RL controller and human gait, specifically: Specifialy, $\delta t_A = t_1 - t_A$, $\delta d_A = d_1 - d_A$, $\delta t_C = t_2 - t_C$, and $\delta d_C = d_2 - d_C$. **On Human Robot Co-adaptation:** 1) The gait duration after online training increases to accommodate inflation/deflation (time it takes to actuate the airbag) as row A) shows that the dashed orange line is longer than solid blue line. 2) Preserved Typical Human Gait: The duration of human stance $d_A, d_C$ changed very little indicating that participants continue walking naturally with their typical gait patterns. 3) Stance Onset Timing $t_1$: as Row B) shows, that for majority of the participants (4 out of 5), RL adjusted the control to align more closely with the human stance onset timing $t_A$ And one participant, the control onset shifted slightly earlier. 4) Swing Onset Timing $t_2$: consistently shifts earlier than the human's actual swing starting time $t_C$, ensuring actuators inflate in time to assist lifting the leg. The offline policy had an even earlier $t_2$, likely due to the attached actuator but the absence of actuator deployment during training. However, if this onset is too early, it could hinder natural knee movement—hence, online learning automatically adjusted it to strike a balance between assistance and preserving gait mechanics.

# I. Incline Task

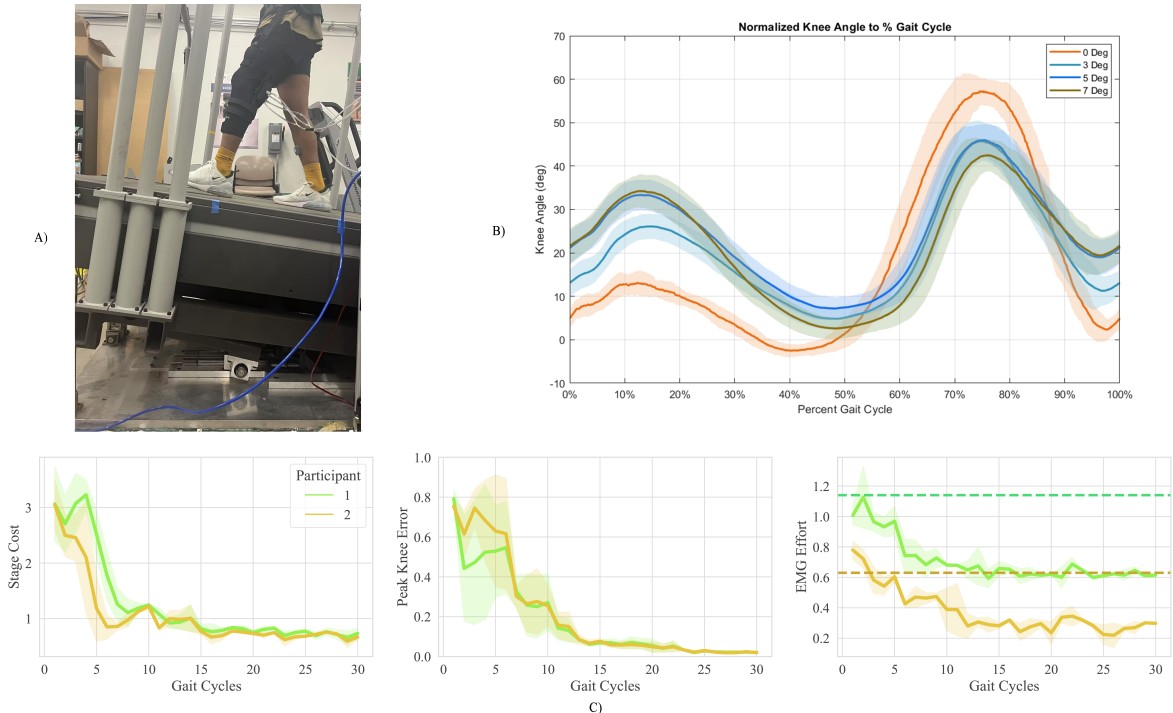

*Figure 10.* A) 7 Degree of Incline Walking B) Walking Profiles vary over incline angles C) Complete Online Training Results of 7 Degree Incline Walking with 2 Participants. The shaded regions represent the 95 % confidence range of the three experiment trials. The x-axis is the number of gaits. The **dashed** line in the EMG effort panel represent the **BASELINE** EMG effort when there is no control of the device.

