# OpenReview forum: "Reinforcement Learning Control of a Physical Robot Device for Assisted Human Walking without a Simulator"
_ICML.cc/2025/Conference — ICML 2025 poster_

### Official Review · Reviewer_QtzX · 2025-02-26

**Overall Recommendation:** 3

**Summary:**

This paper develops an RL application for controlling soft wearable exosuit for human normative walk. Grounded in the motivation that this type of system lacks robust simulators or dynamic models, the paper approaches it from a model-free RL perspective. Furthermore, given the natural lack of data from this problem, the paper leverages the traditional approach of initializing a policy via Imitation Learning from pre-collected trajectories and then applies a RL fine-tune to optimize a notion of costs that encourages safe normative walk and reduced EMG efforts. Based on experiments conducted with five healthy patients, the paper claims that the proposed system improves personalized walking assistance.

**Claims And Evidence:**

- The RIIV “method” improves data quality for learning the exosuit walk controller.

Evidence: The work brings as evidence the prediction error of the controllers learned with the proposed method and two other baselines (the “Direct” method and directly using sensor data). From the bar plots, it is clear that, for some of the control variables, the RIIV method presents better results. Nonetheless, there are some points of concern: first, the presented final training cost (Fig. 2, left) is roughly the same for both cases, which makes unclear why RIIV presents better prediction errors in most variables. Furthermore, none of the results present confidence intervals, so it is unclear if they are statistically significant here.


- Online learning effectively adapted the initial offline policy and provided a personalized control for human-robot normative walk.

Evidence: The presented evidence shows the effect of online RL on top of the offline learned policy, presenting gains in terms of reduced EMG effort and peak knee error across the five participants. While this supports the hypothesis of offline-to-online adaptation, it is still unclear if this controller really brings benefits to humans. To clarify that, I would expect a baseline with no control signal (like a “placebo” controller) and evaluate the performance measures for the patients. The goal is to understand what is the impact of the whole exosuit in the patients. Another important point is: humans also adapt themselves to improve those performance measures, so it is unclear if the raised gains are due to the adaptation of the RL controller OR if they are due to the humans adapting their locomotion. To explain away the latter hypothesis, I would expect a baseline that runs the offline policy for as long as the online phase takes but without any RL adaptation, and after that evaluates the performance measures.

- Human and robot co-adapted to achieve normative walking with reduced EMG effort.

Evidence: The paper provides additional plots presenting the timing and duration of the state/control variables and describes how this evidence supports this co-adaptation hypothesis. Honestly, the text was really unclear and hard to follow, so I believe the paper failed to communicate the findings for this claim. Furthermore, the plots in Figure 4 are hard to read. Lastly, the suggest baseline from the previous point is also important here: it is unclear if the reduced EMG effort is due to the adaptation of the human or the RL controller, or both.

- Both cost terms in the objective are required to achieve stable human-robot normative walk.

Evidence: The paper conduct experiments ablating each term of the objective function to support this claim. While I believe this is the right evidence, the paper struggles to correctly communicate the findings again:it brings to separate figures, one of each representing the ablation of one component. But the figures are separated, and the plot scales are different, which makes really hard to analyze them. Therefore, it is hard to evaluate such a claim. I would suggest to unify Figures 5 and 6, and for each plot, bring 3 different curves: the one with the full objective, and other two ablating each component of the objective.

**Essential References Not Discussed:**

Please see my previous comment.

**Experimental Designs Or Analyses:**

The general experimental design makes sense, although I believe the two baselines mentioned before are key to validate the impact of the proposed system. Besides that, It is not clear how challenging the presented setup is. The experiments are conducted in five healthy patients, in a stable walk of 1 m/s. The problem presents a state-space of 5 dimensions and action-space of 4 dimensions. Thus:

1) It is unclear if the locomotion gaits from these five patients would indeed require a meaningful adaptation (in other words, how different should we expect their locomotion patterns to be?);

2) The state/action spaces are comparable to small benchmarks in RL; while I understand the major challenge comes from the lack of dynamics model and limited data, the presence of expert demonstrations also considerably mitigates that and provides a good base policy for downstream RL.

**Methods And Evaluation Criteria:**

Yes. Since this is an application-driven paper, the evaluation happens directly in the final target application, and the evaluation criteria follows directly from the performance criteria expected from a walk assistance gadget.

**Other Comments Or Suggestions:**

Typos: The work consistently mistakes open quotes.

**Other Strengths And Weaknesses:**

Strengths: I believe the overall problem setting (exosuit control) is really interesting and the proposed methodology fits the problem very well given the lack of a simulator/dynamics model.

Concerns: Besides the concerns previously highlighted, I struggled to find what is the practical impact of the proposed controller. In line with what was described in the Impact Statement, the method (as it is developed and evaluated) is limited to a very restricted locomotion pattern and focuses particularly on healthy patients. While I understand the work provides a proof of concept, I believe it is still very far from a real-world controller, as it would require more extensive evaluation in different locomotion patterns, comprising different speeds, terrains, etc. As an application paper, it would be interesting to describe which applications are the developed controller purposed for, given the content presented in this paper.

**Questions For Authors:**

- How are the safety constraints and tolerance parameters selected? Tables 3, 4, and 5.

- In Appendix G, are the described adjustments to address the OOD issue learned by the controller or is there a human, manual intervention?

**Relation To Broader Scientific Literature:**

I understand the main contribution of the paper as providing a practical RL system for exosuit control under a limited setup (stable 1 m/s walk for healthy humans), which seems to be the first of its kind. The work does bring a very informative Related Work on the area, which helps understand why this is different from exoskeleton control. However, I am not expert on this area and I could not really evaluate if there are other potential methods for exosuit control that should be contextualized to this work.

**Theoretical Claims:**

The work is mostly driven by the application, so the work is more on the empirical side. The work does bring some theoretical analysis, but to the best of my knowledge this is largely inherited from previous work that analyses the RL algorithm properties and not exactly a direct contribution of this work.

---

> ### Author Rebuttal · Authors · 2025-04-01
>
> # We thank the reviewer for thier thoughtful feedback, please check our [new results](https://www.dropbox.com/scl/fo/rgonc4oohtzgf87jqlq3y/AKOxgA5jW9PHt3NKRFLAPcw?rlkey=04igqadzdmyojb9y48zlds5gf&st=g4pizvbd&dl=0)
>
> >Q1 Fig.2 concern
>
> 1)  DIRECT & RIIV are different methods resulting in different physical scales. Therefore, Fig2-left (the learning curve) should only be used to monitor learning convergence, but not for quantitative compassion of performance. For example,  consider only 1 non-zero error in eqn(6) be the swing timing $t_c =0.1$, then in physical time, RIIV error=0.0198s and DIRECT error=0.0946s.  Please note that, the other 4 plots to the right provides more details on compassions of different methods.
> 2)  We have included a revised Fig 4 (now Fig 11). Confidence interval is presented for Fig4-left. Note that, the other plots in Fig. 4 are shown in log scale, which does not reflect physical values of std. We therefore have included a new Table 8 to report actual STDs for each variable.
>
> >Baseline and Isolate co-adapation
>
> 1) We kindly ask the reviewer to refer to Fig 3 and its caption where each participant has a dashed line indicating the baseline EMG effort during natural human walking without exosuit assistance.
> 2) For details on isolating effects of robot and robot adaptation, please refer to Reviewer tqYD Q1.
>
> >Fig 4 concern
>
> We have a new Fig 12. Plz review carefully, especially the caption on coadaptation.
>
> >Fig 5,6 concern
>
> As reviewer suggested, we include a new Fig 13. Since each term in the overall cost differ physically, so are their scales. The results reported here are based on actual values without any pre-/post- process. The purpose of Fig 13 is to show how learning with only a partial objective (EMG or kinematic error alone) fails to guide effective adaptation to achieve normative walking with reduced EMG effort. This is what ablation study is designed for.
>
> > Challenge of the study and Gait veriation
>
> Our problem presents unique and significant challenges that distinguish it from standard simulated tasks:
> 1. Unlike benchmark env based on rigid-body systems with reliable simulators, our system involves a soft exosuit interacting with a human, introducing complex, and time-delayed dynamics. No existing simulator can accurately model this interaction, making our simulator-free, real-world learning approach both necessary and significantly more challenging.
> 2. Benchmark tasks offer well-defined state/action spaces and objective functions. In contrast, our study tackles a first-of-its-kind real-time control problem, where defining meaningful states, actions, and cost functions requires substantial innovation—making problem formulation itself a core contribution of our work.
> 3. Regarding the gait variation, please refer to our detailed response in reviewer Q99a Q3.
> 4. We appreciate the reviewer’s recognition that expert demonstrations help address the data challenge and note that, to our knowledge, no prior work has applied RIIV or this offline-to-online pipeline in soft wearable robotics—making it a novel contribution of our study.
>
> >Complex tasks and real world application
>
> Please refer to our detailed response to reviewer prNk Q3
>
> >Appendix I, centering issue and typos
>
> Figs in Appendix I are discussed in the main paper. Fig 5 and 6 in the ablation study, and Fig 7 in the co-adaptation. Typos will be corrected, thanks.
>
> >Safety Constraints selected
>
> 1. Table 5 presents biomechanics data that define the realistic range of normative walking patterns [Zhang et al., 2020b].
> 2. Based on this reference, the control safety constraints in Table 3 were established. In particular, we account for the actuator’s deflation delay that too long duration may raise safety concern so we cap it to 20% gait phases. The target states used in our controller correspond to the most commonly observed values within normative walking.
> 3. Table 4 outlines constraints in the RL framework to ensure participants maintain normative walking and are not exposed to unsafe or misleading actuator behavior.
>
> >human, manual intervention of OOD
>
> There was no human intervention or cherry-picking throughout the study—all results reflect true experimental data from all five participants. The adaptations were learned automatically by the RL controller during online training. In the offline phase, the exosuit was inactive, and data were collected from unassisted walking, meaning no actuator-induced effects were present. As shown in Figure 9A–B, activating the actuators led all participants to adjust their gait, introducing a distribution shift and potential OOD issue. Through online learning, the RL policy adapted to this shift as part of a natural human-exosuit co-adaptation process, without manual tweaking. As demonstrated in Fig 12, this adaptation was crucial for real actuator behavior and achieving normative gait patterns with reduced muscular effort.

---

> > ### Comment · Reviewer_QtzX · 2025-04-04
> >
> > Thanks for your rebuttal. In light of the new evidence, I decided to increase my score from 2 to 3.

---

> > > ### Author Response · Authors · 2025-04-04
> > >
> > > Thank you! and we appreciate your evaluation and the opportunity to improve our work  based on your feedback.

---

### Official Review · Reviewer_prNk · 2025-03-09

**Overall Recommendation:** 2

**Summary:**

This paper presents an innovative approach to controlling soft exosuits for assisted human walking using reinforcement learning (RL) without relying on a simulator. The authors propose an online Adaptation from an offline Imitating Expert Policy (AIP) approach that addresses key challenges in RL-based control of physical devices, including limited data, absence of simulators for human-robot interaction, computational constraints, and the need for personalized control while ensuring safety. The method first uses offline learning to mimic human expert actions through real walking demonstrations without robot assistance, then initializes online actor-critic learning to optimize personalized robot assistance. The authors demonstrate their framework on five human participants, showing consistent performance improvements in terms of reduced human effort while maintaining normative walking patterns. The paper also provides qualitative performance guarantees for their online RL method, including learning convergence, dynamic stability, and solution optimality.

**Claims And Evidence:**

The claims made in the submission are well-supported by evidence. The authors claim that their AIP approach can effectively control a soft exosuit to assist human walking without a simulator, which is demonstrated through experiments with five human participants. The claim that online adaptation can personalize the control for individual users is supported by the consistent reduction in EMG effort (muscle activity) across all participants after online training, despite initial variations. The paper also claims that their data-centric approach is more effective than algorithm-centric approaches for this application, which is supported by comparing their RIIV (reducing intra-person and inter-person variation) method with a benchmark direct normalization approach, showing improved action divergence and better alignment with ground truth data.

**Essential References Not Discussed:**

N/A

**Experimental Designs Or Analyses:**

The experimental design is robust and well-executed. The authors test their approach on five human participants walking on a treadmill at a constant speed, with clear safety constraints to ensure participant comfort and safety. The experiments include both offline data collection using a Motion Capture system for ground truth and online learning using IMU sensors for real-time control. The analyses include quantitative comparisons of performance metrics before and after online learning, as well as ablation studies to understand the contribution of different components of the cost function. The results are presented clearly with appropriate statistical measures, and the authors do not selectively present results, which enhances the credibility of their findings.

**Methods And Evaluation Criteria:**

The proposed methods and evaluation criteria are appropriate for the problem at hand. The authors use a combination of offline imitation learning and online reinforcement learning, specifically direct heuristic dynamic programming (dHDP), which is well-suited for the constraints of real-time control with physical devices. The evaluation criteria include stage cost, peak knee error, and EMG effort, which effectively capture both the quality of the walking assistance (normative walking) and the reduction in human effort. The authors also consider important practical aspects such as time to convergence of RL online learning and action divergence to measure offline policy optimality. The experimental design involving five human participants provides a reasonable sample size for demonstrating the effectiveness and generalizability of the approach.

**Other Comments Or Suggestions:**

Other Comments Or Suggestions
- The paper would benefit from a more detailed discussion of the limitations of the approach and potential directions for future work.
- A more comprehensive analysis of the computational requirements and real-time performance of the proposed method would be valuable for understanding its practical applicability.
- The authors could consider exploring the potential of transfer learning between participants to further reduce the online learning time for new users.
- The paper mentions that the approach could be applied to other wearable robotics applications, but does not elaborate on how the method might be adapted for different types of devices or assistance tasks.

**Other Strengths And Weaknesses:**

Strengths:
- The paper addresses a significant practical challenge in controlling soft exosuits for human assistance, with potential real-world impact in rehabilitation and assistive technologies.
- The approach is pragmatic, focusing on data quality rather than algorithm complexity, which is particularly valuable for applications with limited data and computational resources.
- The experimental validation on five human participants demonstrates the robustness and generalizability of the approach.
- The paper provides both theoretical guarantees and empirical evidence for the effectiveness of the proposed method.

Weaknesses:
- The paper could benefit from a more detailed comparison with state-of-the-art methods in exosuit control, beyond the ablation studies presented.
- The long-term effects of the proposed control approach on human adaptation and learning are not explored, which would be valuable for understanding the practical implications of the technology.
- The paper focuses primarily on level ground walking, and it's unclear how well the approach would generalize to more complex locomotion tasks such as stair climbing or uneven terrain navigation.
- The sample size of five participants, while reasonable for a proof-of-concept, may not be sufficient to capture the full range of human variability in gait patterns and responses to exosuit assistance.

**Questions For Authors:**

- How does the computational complexity of your online learning approach compare to other RL methods, and what are the implications for real-time control on embedded systems with limited resources?
- Your experiments focus on level ground walking at a constant speed. How do you envision extending your approach to more complex locomotion tasks or varying walking speeds, and what additional challenges might arise?
- The paper demonstrates co-adaptation between the human and the robot during online learning. Have you observed any longer-term adaptation effects in human participants after extended use of the exosuit with your control approach, and how might these effects influence the design of adaptive controllers for wearable robots?

**Relation To Broader Scientific Literature:**

The paper effectively situates its contributions within the broader scientific literature on soft exosuit control, reinforcement learning for robot control, and offline-to-online RL approaches. The authors acknowledge prior work on human-in-the-loop optimization methods for exosuit control and highlight the limitations of these approaches in terms of requiring robust dynamical models and lacking online tuning capabilities. They also discuss recent advances in offline-to-online RL and note that most existing approaches rely on extensive simulations or vast amounts of offline data, which are not available in their application. The paper's data-centric approach represents a novel contribution to the field, addressing a gap in the literature on RL control of physical devices without simulators.

**Theoretical Claims:**

The paper makes several theoretical claims about the properties of their learning process and control performance, including learning convergence, solution optimality, and control system stability. While the main paper provides a high-level overview of these claims, the authors reference detailed proofs in the appendices. The theoretical framework appears sound, building on established principles from reinforcement learning and control theory. The authors acknowledge the challenges of controlling soft inflatable actuators due to their nonlinear nature, material properties, and manufacturing variations, and provide a theoretical analysis that accounts for these factors.

---

> ### Author Rebuttal · Authors · 2025-04-01
>
> # We thank the reviewer for thier thoughtful feedback, please check our [new results](https://www.dropbox.com/scl/fo/rgonc4oohtzgf87jqlq3y/AKOxgA5jW9PHt3NKRFLAPcw?rlkey=04igqadzdmyojb9y48zlds5gf&st=g4pizvbd&dl=0)
>
> >Q1 Compare with sota method
>
> We thank the reviewer for the suggestion:
> 1. We have added Table 9, which summarizes related work that uses EMG as a performance metric. Our RL-based method achieves an average EMG reduction of 20%, significantly outperforming rule-based methods such as the one reported by Sridar, which achieved only 7.37% reduction.
> 2. The model based method and Bayesian optimization required probabilistic models or approxiamation of the objective which
> is not suitable for our problem.
> 3. Please refer to Reviewer Q99a Q1 for further detailed explainations.
>
> >Q2 Long Term effects
>
> As the application is the first study of its kind, long term study falls outside the scope of this study. However:
> 1. A potential challenge for long-term deployment is actuator timing drift due to wear and tear. In our study, all five participants used the
> same exosuit hardware across sessions, so any natural degradation—such as material fatigue or delay shifts—was implicitly
> captured during learning. In such effects occurred, AIP would have adapted to them as part of the online training process
> 2.  As shown in the newly added Fig 9 C), the AIP policy consistently maintains reduced EMG effort across all evaluation sessions with minimal variation.
> 3. Please refer to Reviewer Q99a Q2 for related comments.
>
> >Q3 Generalize and challenges of more complex tasks
> 1. As the first study of its kind, our work focuses on level-ground walking at a constant speed—a standard and widely adopted practice in the wearable robotics community. As shown in Table 9 and 10, related studies have also primarily evaluated their methods under similar conditions. This controlled and repeatable environment serves as a foundational step for developing and validating the proposed AIP method before extending it to more complex locomotion tasks.
> 2. Regarding generalization of AIP, we introduced new results involving treadmill walking on a 7-degree incline, where the AIP method continues to perform robustly. Please refer to Reviewer tqYD, Q4 for related comments.
> 3. Real-world locomotion includes diverse tasks such as stair climbing and uneven terrain. While the core AIP principle remains applicable, these scenarios introduce challenges such as greater variability in human motion, the need for terrain-aware sensing, and potentially longer training times due to increased dynamic complexity.
>
> >Q4 Apply to other applications
>
> 1. The formulation of the state, action, and cost need to be adapted for different tasks.
> 2. For RIIV, captures the invariant feature of application and can potential inprove sample efficency. In other human-robot interactive systems, the principles of RIIV should still apply to identifying invariant task-related characteristics.
> 3. To extract the offline policy, we used a motion capture system commonly adopted in human-robot locomotion studies.
>
> >Q5 Gait pattern varibility
>
> Please refer to Reviewer Q99a Q3 for related comments.
>
> >Q6 Limitation and Transfer learning
>
> 1. Our method currently controls a single leg. Extending to bilateral control introduces challenges in coordinating and synchronizing assistance across limbs. Multi-agent RL offers a promising direction to address this open question.
> 2. Data efficiency is a known limitation of RL. We agree that transfer learning is a promising direction and we are actively exploring this approach, as it can accelerate convergence and reduce user-specific data needs.
>
> >Q7 Computational Complexity
>
> 1. The computational complexity of our online learning method is similar to that of DDPG, as both use the vanilla deterministic policy gradient framework. Our actor and critic networks have two fully connected layers with 256 hidden neurons, and a small batch size of 5 is used during online training. Based on the complexity estimate $O( Batch * layer * neuron^2)$, the method remains lightweight and efficient.
>
> 2. RIIV is a data-centric method with low computational complexity of $O(state)$, designed to minimize unnecessary input variance. This alignment improves data efficiency (Fig 2) and supports stable learning with compact networks and small batch sizes—without compromising performance.
>
> 3. For real-time deployment, we use a Raspberry Pi 5 microcontroller, with power consumption ranging from 2.7 to 12 watts.  It has consistently supported both control and inference in real time across all five participants, with no observed performance issues.

---

### Official Review · Reviewer_Q99a · 2025-03-12

**Overall Recommendation:** 4

**Summary:**

This paper presents an RL-based control framework for a soft exosuit that assists human walking without a simulator. The proposed Adaptation from an offline Imitating Expert Policy (AIP) approach learns from human walking demonstrations and refines control using dHDP. AIP prioritizes data quality over large-scale simulations, enabling real-time personalized assistance. Tested on five participants, the method reduces muscle effort while maintaining normative walking patterns, demonstrating learning convergence, stability, and optimality.

## update after rebuttal
Thank you for the thoughtful rebuttal. I appreciate the effort to address my concerns. I will maintain my original score.

**Claims And Evidence:**

Yes.

**Essential References Not Discussed:**

N/A

**Experimental Designs Or Analyses:**

Yes. The experimental designs are reasonable, however, it would be stronger if direct comparisons to alternative methods were provided, such as human-in-the-loop optimization, traditional rule-based control strategies, or a simpler heuristic baseline. Also, the experiment setup is simple now, it would be beneficial to explore how AIP adapts over extended use.

**Methods And Evaluation Criteria:**

Yes.

**Other Comments Or Suggestions:**

N/A

**Other Strengths And Weaknesses:**

Strengths:
1. Simulator-free RL for exosuit control and offline-to-online. The proposed AIP framework eliminates the need for a simulator, addressing a major data limitation in prior RL-based exosuit control methods. The combination of offline imitation learning and online RL adaptation enables personalized exosuit assistance.
2. Data-centric approach. This paper introduces Reducing Intra- and Inter-Person Variation (RIIV) to improve data quality for imitation learning.
3. Convincing results. The experimental results demonstrate reduced muscle effort for human participants and fast adaptation.

Weaknesses:
1. Unclear long-term adaptation**.** The paper does not explore whether AIP remains stable over prolonged use or how well it generalizes to real-world walking conditions beyond a treadmill setup.
2. Limited participant diversity – The study only includes five participants, primarily young, fit females and heavier males, which limits the generalizability of the findings. Older individuals, lighter males, and a broader range of body compositions could exhibit different gait patterns and adaptation responses, affecting the model’s robustness.

**Questions For Authors:**

1. What are the computational requirements for the real-time deployment? Could the method work on low-power edge devices?
2. How stable is the method over multiple days of use? Does the learned policy continue improving over time, or does it plateau?
3. What are the biggest challenges in scaling your method to a larger, more diverse population?
4. Did you collect any subjective feedback from participants regarding their comfort, ease of movement, or perceived assistance from the exosuit? If so, what were their suggestions for improvement?

**Relation To Broader Scientific Literature:**

This paper builds on prior work in wearable robotics, RL, and human-in-the-loop optimization, addressing the unique control challenges of soft exosuits. Prior RL-based exosuit control relied on sim-to-real transfer, whereas this paper eliminates the need for a simulator by combining offline imitation learning with online adaptation.

**Theoretical Claims:**

Yes. Section F. dHDP solution and properties in the supplementary material.

---

> ### Author Rebuttal · Authors · 2025-04-01
>
> # We thank the reviewer for thier thoughtful feedback, please check our [new results](https://www.dropbox.com/scl/fo/rgonc4oohtzgf87jqlq3y/AKOxgA5jW9PHt3NKRFLAPcw?rlkey=04igqadzdmyojb9y48zlds5gf&st=g4pizvbd&dl=0)
> >Q1 Compare with other methods
>
> We thank the reviewer for the suggestion. In response, we have added a comparison Table 9 summarizing related work that
> uses EMG as a performance metric:
> 1. There is no simple intuitive heuristic control and as shown in the Table 9, rule-based methods consistently underperform compared to automatic control approaches. This is because they rely on expert-defined heuristics and lack adaptability to individual variations. The only known method for knee assistance, proposed by Sridar et al., requires manual tuning and achieves only a modest 7.37% EMG reduction. In contrast, our AIP method achieves an average 20% reduction in EMG, requires significantly less time, and learns automatically through reinforcement learning.
> 2. Some related approaches, such as Bayesian optimization (BO), rely on probabilistic models or approximations of the objective function. These introduce early approximation errors and struggle to extrapolate effective policies from the limited state-action data available in our study. Furthermore, model-based methods are generally impractical for soft exosuits due to the difficulty of accurately modeling their highly compliant and nonlinear dynamics.
> 3. This further motivates our use of RL, as it enables us to collect human data for offline learning and extract an initial policy that can be used to effectively initialize online adaptation.
>
> >Q2 Extended Use Issue
>
> 1. As this is the first application of its kind, long-term testing across multiple days or in uncontrolled, real-world environments falls outside the scope of this study—consistent with prior work in the field (see Tables 9 and 10).
> 2. However our experimental design offers insight into stability over extended use. Each participant completed six cycles of 10 minutes walking followed by 5 minutes rest, totaling 7.5 hours for all participants. Refer to newly added Fig 9C, the AIP policy consistently reduced EMG effort with minimal variation, suggesting stable performance post-training.
> 3. We expect AIP to maintain performance once converged. Potential timing drift due to actuator wear is inherently accounted for, as all five participants used the same hardware. Any degradation would have been captured during training. Moreover, our pneumatic actuators are easily replaceable, supporting routine maintenance and long-term deployment.
> 4. Generalization to real-world conditions aligns more with industrial-scale evaluation. While outside the current scope, we believe AIP’s model-free, data-driven approach can generalize effectively, provided additional training data is collected in those settings.
>
> >Q3 Participant and Gait Diversity
> 1. Our participant pool included both males (3) and females (2), with a weight distribution of 69 ± 11.6 kg and height of 1.63 ± 0.09 m, demonstrating equal or greater variability in key physical characteristics compared to related studies in Table 10.
> 2. As shown in the newly added Fig 10, we present a histogram illustrating the range of gait patterns observed in our study. Based on established tolerance bounds from prior bio literature, our participants’ gait patterns span a broad spectrum within normative walking ranges. This indicates that our study captures a wide range of commonly observed gait variations.
> 3. Additionally, since this is the first study of its kind—with no prior precedent for the proposed method—IRB approval was essential to ensure participant safety. Including individuals with extremely diverse body types could introduce safety concerns, such as improper fit or reduced actuator effectiveness.
> 4. To further examine gait diversity and robustness, we introduced new results involving treadmill walking on a 7-degree incline, where the AIP method continues to perform robustly. For details, please refer to our response to Reviewer tqYD, Q4.
>
> >Q4 Computational Requirements
>
> Yes, our method runs effectively on low-power edge devices. For real-time deployment, we use a Raspberry Pi 5 microcontroller, with power consumption ranging from 2.7 to 12 watts. For details, please refer to our response to Reviewer prNK, Q7.
>
> >Q5 Subject Feedback
>
> 1. During training, we asked participants every five steps, “Do you understand the soft suit’s behavior?”
> 2. Participants’ awareness improved gradually. Within the first 20 steps, they commonly responded, “I don’t understand” By 50 steps, they reported, “I kind of know” and by 100 steps, most expressed, “I am fully aware”.
> 3. There are two directions to improve this process. First, enhancing the sample efficiency could reduce the initial "confusion" phase. Second, adding real-time visual feedback on when and how assistance is being applied—could help participants understand suit’s behavior more quickly.

---

### Official Review · Reviewer_tqYD · 2025-03-14

**Overall Recommendation:** 4

**Summary:**

This paper presents AIP (Adaptation from an offline Imitating expert Policy) for controlling a soft inflatable exosuit to assist human walking without relying on a simulator. The approach first learns from human walking demonstrations (offline phase), then adapts this policy online to personalize assistance. The method is validated on five participants, demonstrating reduced muscle effort while maintaining normative walking patterns.

## update after rebuttal
I acknowledge the authors' rebuttal and maintain my original assessment.

**Claims And Evidence:**

*Key Claim:* AIP enables effective control of soft exosuits without a simulator

*Evidence:* The authors demonstrate successful implementation on five participants with consistent performance improvement. Their approach successfully adapts to individual differences without requiring a complex simulator - which would be difficult to develop for this application due to the unpredictable human-soft robot interaction dynamics.

**Essential References Not Discussed:**

NA

**Experimental Designs Or Analyses:**

*Offline-to-online transition methodology:* The authors' approach of using a single participant's data for offline learning and then testing generalization to other participants is valid for demonstrating adaptability. The experimental protocol included appropriate warm-up periods and rest periods between sessions to minimize fatigue effects.

*Baseline comparisons:* The comparison against previous reported EMG reductions (7.37%) provides context for their 20% reduction. However, a within-study baseline comparison showing human adaptation alone would have strengthened the experimental design by better isolating the benefits of the co-adaptation process.

**Methods And Evaluation Criteria:**

The offline-to-online approach with RIIV data processing addresses the key challenge of working without simulation in human-robot learning. Performance metrics comprehensively capture both technical performance (stage cost, convergence time) and human factors (EMG reduction, knee error). The participant pool of five individuals provides reasonable diversity for initial validation.

This work would be a good reference for practical reinforcement learning for physical human-robot interaction, especially in scenarios where accurate simulation and large expert data collection is infeasible.

**Other Comments Or Suggestions:**

NA

**Other Strengths And Weaknesses:**

**Strengths**
- Addresses the critical challenge of developing controllers for soft exosuits without simulators, which are nearly impossible to create for these complex human-robot systems.
- Demonstrates successful direct learning in a physical environment with human subjects, a significant achievement in reinforcement learning.
- Shows robust personalization across participants with measurable benefits (20% EMG reduction), outperforming previous methods (~7.37% reduction reported in comparable studies).
- The participant diversity (varying height, weight, gender) is appropriate and exceeds similar studies in the field.
Includes supplementary videos demonstrating walking with the exosuit, strengthening the empirical validation.

**Weaknesses**
- The method relies on established RL algorithms (BC and dHDP) rather than introducing algorithmic innovations, though the paper's true contribution is in the practical implementation rather than novel algorithms.
- The ablation study could provide deeper insight into the specific conditions under which simulator-free learning becomes advantageous versus when simulations might still be beneficial.
- While the authors address population diversity adequately for this study, a more explicit roadmap for extending to more complex scenarios (variable walking speeds, different terrains) would strengthen future applications.

**Questions For Authors:**

NA

**Relation To Broader Scientific Literature:**

NA

**Theoretical Claims:**

NA

---

> ### Author Rebuttal · Authors · 2025-03-31
>
> # We thank the reviewer for thier thoughtful feedback, please check our [new results](https://www.dropbox.com/scl/fo/rgonc4oohtzgf87jqlq3y/AKOxgA5jW9PHt3NKRFLAPcw?rlkey=04igqadzdmyojb9y48zlds5gf&st=g4pizvbd&dl=0)
> >Q1 Isolating the benefit of co-adaptation
>
> To address the issue of isolating the benefits of co-adaptation, please refer to Fig 4, 7, and the newly added results (Fig 9) from the same previous experiment data.
> 1. human-robot co-adaptation is coupled as the human brain sensorimotor system governs their behavior during robot assisted walking. Isolating human adaptation is not feasible for the current study. But human adaptation reflected by human walking behavior naturally emerges in response to robot learning control.
> 2. To isolate the effect of co-adaptation, the only thing we could do was to disable the robot’s adaptation by freezing the policy updates—this setup is shown in Fig 9.
> 3. For Fig 9, in row A (baseline), participants walked naturally and showed their natural gait pattern; in row C (after online training), the policy is already optimized, so no further human adaptation is needed. In both cases, we observe no sign of human adaptation.
> 4. In row B (learned offline policy), the Offline policy is not optimal and needs to be personalized. This is where we speculate that the participants may recognize the mismatch and attempt to adapt, but since there is no clear or consistent direction, the participants’ responses appeared without clear pattern or trend. For example, P1 shows delayed timing in $t_A$, while others do not exhibit this behavior. Similarly, P5 demonstrated a slightly earlier duration in  $d_C$, which was not seen in other participants.
> 5. Evidence of co-adaptation during online learning was clear. Fig 4 and 7 further support this: humans slightly adjust gait duration to accommodate actuator inflation/deflation, while the RL controller adapts its onset timing. This coordinated adaptation is evident in the bar plots of Fig 4 (revised presentation is now Fig 12), showing significant changes in timing and duration between human gait kinematics and robot control.
>
> >Q2 paper’s true contribution is in the practical implementation rather than novel algorithms.
>
> Yes, the offline-to-online learning framework is well-established in the robotics community. Additionally, our proposed RIIV is not limited to the specific locomotion task presented in this study. The core principles of RIIV are broadly applicable to other human–robot locomotion scenarios (For example, our newly added experimental data involves walking on an inclined treadmill at 7 degrees, please refer to Fig 8).
>
> >Q3 when simulator-free learning becomes advantageous VS simulatior-based learning
>
> 1. We thank the reviewer for this excellent question. In our case, a simulator is not available and therefore, it is not feasible for us to  quantitatively compare simulator-based and simulator-free approaches in this study.
> 2. However, to shed some light, we may consider minimum fidelity requirements for simulations. A simulator can be beneficial if it meets key fidelity criteria—such as accurately replicating system dynamics and kinematics. The feasibility and usefulness of simulation, however, depend heavily on the application. For example, a cart-pole task requires much simpler modeling than a human-in-the-loop soft exosuit system.
> 3. Even if a model were available to simulate our system’s dynamics, deploying it for real-time control would remain highly challenging as in the well-known sim-to-real gap in reinforcement learning
> 4. In contrast, our simulator-free approach directly engages with the real-world complexities without relying on approximations. However, it comes with its own challenges—particularly the need for careful problem formulation and data-centric design (e.g., our RIIV method) to ensure efficient and effective learning while capturing the invariant dynamics of the human-exosuit interaction.
> 5. Ultimately, the advantage of simulator-free learning lies in its realism and adaptability, while the benefit of simulation
> depends on the availability of a high-fidelity model. In our context, simulator-free learning is a practical and effective choice, and perhaps, the only choice now.
>
> >Q4 Roadmap for more complex scenarios
> 1. We agree and our current roadmap focuses on generalizing the method beyond basic treadmill simulated level-ground walking  to include more complex and realistic tasks and scenarios, including walking at variable speeds, inclined and declined walking, among others.
> 2. As a first step of extension, we have begun developing/validating the approach on inclined walking. Please refer to the newly added Figure 8, which shows results from a 7-degree incline treadmill walking. As shown,  increasing the incline significantly alters gait kinematics. However, as shown in Panel C, our AIP method remains robust, consistently reducing EMG effort below BASELINE levels across all participants.

---

### Decision · Program_Chairs · 2025-05-01

**Decision:**

Accept (poster)

**Comment:**

This is a resubmission improved based on ICLR reviews. The AC agrees with the majority of reviewers that the quality is ok after substantial improvements. Some concerns still exist, e.g. whether the topic is a broad fit for ML community or assistive robotics community.